# Canonic Signed Spike Coding
# for Efficient Spiking Neural Networks

## Abstract

Spiking Neural Networks (SNNs) seek to mimic the spiking behavior of biological neurons and are expected to play a key role in the advancement of neural computing and artificial intelligence. The conversion of Artificial Neural Networks (ANNs) to SNNs is the most widely used training method, which ensures that the resulting SNNs perform comparably to ANNs on large-scale datasets. The efficiency of these conversion-based SNNs is often determined by the neural coding schemes. Current schemes typically use spike count or timing for encoding, which is linearly related to ANN activations and increases the required number of time steps. To address this limitation, we propose a novel Canonic Signed Spike (CSS) coding scheme. This method incorporates non-linearity into the encoding process by weighting spikes at each step of neural computation, thereby increasing the information encoded in spikes. We identify the temporal coupling phenomenon arising from weighted spikes and introduce negative spikes along with a Ternary Self-Amplifying (TSA) neuron model to mitigate the issue. A one-step silent period is implemented during neural computation, achieving high accuracy with low latency. We apply the proposed methods to directly convert full-precision ANNs and evaluate performance on CIFAR-10 and ImageNet datasets. Our experimental results demonstrate that the CSS coding scheme effectively compresses time steps for coding and reduces inference latency with minimal conversion loss.

## 1 Introduction

Spiking Neural Networks (SNNs), recognized as the third generation of neural network models, are inspired by the biological structure and functionality of the brain (Wang et al., 2020). Unlike traditional Artificial Neural Networks (ANNs), which rely on continuous activation functions, SNNs utilize discrete spiking events. This enables SNNs to capture temporal dynamics and process information in a manner that closely resembles brain activity (Taherkhani et al., 2020). The event-driven nature of SNNs aligns with the brain's energy-efficient computational paradigm, offering potential for more efficient and low-power computing systems (Yamazaki et al., 2022).

The two primary learning algorithms for SNNs are gradient-based optimization and ANN-SNN conversion. Directly training using supervised backpropagation is challenging due to the non-differentiable nature of spike generation (Lee et al., 2020; 2016). The conversion-based method, however, offers a practical approach to overcome this difficulty and has produced the best-performing SNNs (Deng & Gu, 2021; Bu et al., 2022; Ding et al., 2021).

Encoding the ANN activations into spike trains is a prerequisite for successful ANN-SNN conversion. Various coding schemes, such as rate coding and temporal coding, have been proposed to describe neural activity (Guo et al., 2021). Rate coding maps the number of spikes to the corresponding ANN activation (Cao et al., 2015). In contrast, temporal coding focuses on the precise timing or patterns of spikes (Rueckauer & Liu, 2018; Kim et al., 2018; Han & Roy, 2020). For example, Time-to-First-Spike (TTFS) coding maps the the activation value to the time elapsed before the first spike (Stanojevic et al., 2022).

However, both the spike counts in rate coding and the spike timing in TTFS coding are linearly related to the encoded activation. This necessitates a large number of time steps to provide sufficient encoding granularity (Stanojevic et al., 2023; Meng et al., 2022). Recent works have proposed alleviating these problems by quantizing the ANN activations before conversion (Hu et al., 2023;

Bu et al., 2023; Hao et al., 2023). This approach simplifies the encoding process but introduces additional quantizing and training overhead. Our goal is to develop a novel encoding method that can directly convert full-precision ANNs while reducing the number of time steps required.

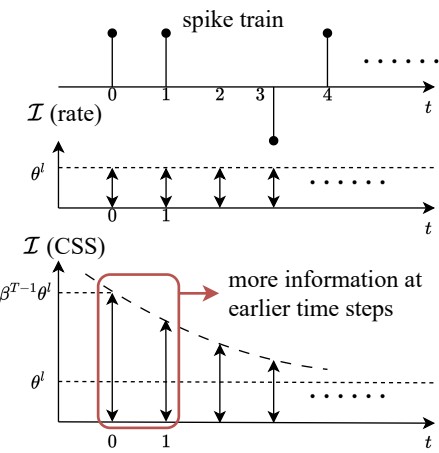

Figure 1: Different interpretations of the same spike sequence. $\mathcal{I}$ denotes the information encoded in a spike. $\theta^l$ is the spike amplitude. $\beta$ is the membrane potential amplification coefficient and $T$ is the total length of the sequence.

In the study of the temporal information dynamics of spikes, Kim et al. (2022) found that after training, information becomes highly concentrated in the first few time steps. This observation led us to hypothesize that the spikes at earlier time steps carry more information and contribute more to the membrane potential. Consequently, by gradually amplifying the membrane potential over time, we increase the influence of earlier spikes. This mechanism essentially assigns exponentially decreasing weights to a spike sequence, with the smallest weight being one (applied to the final spike). This results in a significant enhancement in the encoding capacity of the spike sequence for a given length. Due to this fixed weight pattern during neural computation, we refer to these spikes as canonical.

However, we observed that after weighting, spikes tend to concentrate in later time steps. This phenomenon occurs because earlier spikes now encode larger values, making them less likely to be fired after stimulation. As the spatial depth increases, the spike distribution becomes more biased toward later time steps, leading to significant performance degradation. We refer to this phenomenon as temporal coupling of weighted spikes.

To mitigate this, we introduce negative spikes and lower the firing threshold of neurons to promote earlier spike emission. Neurons are also equipped with a negative spike threshold, allowing them to generate negative spikes that correct excessive firing. This combination results in the Canonic Signed Spike (CSS) coding scheme and the Ternary Self-Amplifying (TSA) neuron model. To better balance the trade-off between coding time steps and inference latency in CSS coding, we introduce a one-step silent period into the TSA neuron, which improves both performance and efficiency of the resulting SNN.

The main contributions of this paper can be summarized as follows:

- By assigning weights to the spikes, we introduce non-linearity into the coding process and compress the time steps to a logarithmic scale. Neurons amplify the membrane potential at each time step, thereby obtaining more information from the preceding spikes.

- We find that weighted spikes are prone to temporal coupling during neural computation, presenting the biggest challenge when incorporating non-linearity in spike coding. We analyze the underlying reasons and introduces negative spikes along with the TSA neuron model to address this issue.

- We demonstrate the effectiveness of the CSS coding scheme on the CIFAR-10 and ImageNet datasets. The results show that the proposed method effectively reduces both the required coding time steps and inference latency. Additionally, the CSS coding scheme offers energy efficiency advantages over both rate coding and temporal coding.

## 2 RELATED WORK

Currently, the mainstream coding schemes in converted SNNs are rate coding and TTFS coding. Rate coding represents different activities with the number of spikes emitted within a specific time window. Early research efforts focused on reducing conversion loss, leading to methods such as weight normalization (Diehl et al., 2015), threshold rescaling (Sengupta et al., 2019), and soft-reset neuron models (Han et al., 2020). More recent work has shifted towards reducing the number of time steps by optimizing neuron parameters. Meng et al. (2022) introduced the Threshold Tuning

and Residual Block Restructuring (TTRBR) method to minimize conversion error in ResNet architectures with fewer time steps. Bu et al. (2022) proposed optimizing the initial membrane potential to reduce conversion loss when using a small number of time steps.

Despite these optimizations, deep networks or large datasets still require hundreds of time steps to achieve satisfactory results. To address this, recent works in rate coding have explored quantizing the ANNs before conversion (Hao et al., 2023; Bu et al., 2023; Hu et al., 2023). This approach directly reduces the number of activations that need to be mapped, providing an alternative way to minimize time steps. Notably, this approach is complementary to ours. The proposed encoding scheme can also convert quantized ANNs and further reduce the required number of time steps.

Due to the functional similarity to the biological neural network, SNNs are highly compatible with temporal coding. Rueckauer & Liu (2018) were the first to attempt converting an ANN to a TTFS-based SNN. While this coding method significantly increased sparsity by limiting each neuron to fire at most one spike, they observed large conversion errors, even on MNIST dataset. Stanojevic et al. (2022) demonstrated that an exact mapping from ANN to TTFS-based SNN is feasible but needs hundreds of time steps for accurate encoding. Yang et al. (2023) proposed a TTFS-based conversion algorithm with dynamic neuron threshold and weight regularization. They completed the conversion with 50 time steps per layer. Despite the reduction in the number of time steps per layer, TTFS coding still suffered from high output latency in deep networks for its layerwise processing manner. Han & Roy (2020) introduced the Temporal-Switch-Coding (TSC) scheme, where each input pixel is represented by two spikes, and the time interval between them encodes pixel intensity. However, as this time interval remains linearly related to activation, the issue of long latency persists.

Some recent works have also incorporated non-linearity into the coding process. Stöckl & Maass (2021) and Rueckauer & Liu (2021) used spikes to encode the "1"s in the binary representations of ANN activations. However, both works did not address the temporal coupling issue caused by weighted spikes. Instead, they adopted an approach similar to TTFS coding, where neurons must wait for the arrival of all input spikes before firing. In contrast, our approach facilitates the greatest extent of synchronous neural computation, thereby reducing both the coding time steps and output latency.

## 3 PRELIMINARIES

### 3.1 SPIKING NEURONS

Spiking neurons communicate through spike trains and are interconnected via synaptic weights. Each incoming spike contributes to the postsynaptic neuron's membrane potential, and a spike is generated when the potential reaches a predefined threshold. Generally, a spike sequence $S_i^l[t]$ in the SNN can be expressed as follows:

$$S_i^l[t] = \sum_{t_i^{l,f} \in \mathbb{F}_i^l} \theta^l \delta[t - t_i^{l,f}]$$

(1)

where $i$ is the neuron index, $l$ is the layer index, $\theta^l$ is the spike amplitude, $\delta[\cdot]$ denotes an unit impulse[1], $f$ is the spike index, and $\mathbb{F}_i^l$ denotes a set of spike times which satisfies the firing condition:

$$t_i^{l,f} : o_i^l[t_i^{l,f}] \geq \theta^l$$

(2)

where $o_i^l[t]$ denotes the membrane potential before firing a spike. Conversion-based works often employ soft-reset IF neuron model, where the membrane potential is subtracted by an amount equal to the spike amplitude for reset. Specifically, its dynamics can be expressed as follows:

$$u_i^l[t] = u_i^l[t-1] + z_i^l[t] - S_i^l[t]$$

(3)

where $u_i^l[t]$ denotes the membrane potential after firing a spike and $z_i^l[t]$ denotes the integrated inputs:

$$z_i^l[t] = \sum_j w_{ij}^l S_j^{l-1}[t] + b_i^l$$

(4)

where $w_{ij}^l$ is the synaptic weight and $b_i^l$ is the bias. For clarity, definitions of the common symbols are provided in Table 1.

---

[1] $\delta[t]$ takes the value 1 at $t = 0$, and 0 otherwise

Table 1: Common symbols in this paper.

| Symbol | Definition | | Symbol | Definition |
|--------|-----------|---|--------|-----------|
| $l$ | Layer index | | $\beta$ | Amplification factor |
| $i, j$ | Neuron index | | $w_{ij}^l$ | SNN weight |
| $S_i^l[t]$ | Spike sequence | | $\hat{w}_{ij}^l$ | ANN weight |
| $o_i^l[t]$ | Membrane potential before firing | | $b_i^l$ | SNN bias |
| $u_i^l[t]$ | Membrane potential after firing | | $\hat{b}_i^l$ | ANN bias |
| $z_i^l[t]$ | Integrated inputs (PSP)[1] | | $T$ | Time steps for coding |
| $\theta^l$ | Spike amplitude | | $\bar{\theta}^l$ | Initial spike amplitude |

[1] Postsynaptic potential

## 3.2 ANN-SNN CONVERSION

The ANN-SNN conversion typically involves the following two key steps: 1) selecting an appropriate encoding method to represent ANN activations as spike trains, and 2) adopting a suitable neuron model that ensures the generated spike trains accurately encode the outputs of the corresponding ANN neurons. Note that this process results from the joint effect of the encoding scheme and the neuron model.

The most widely used and State-Of-The-Art (SOTA) approaches employ (signed) soft-reset IF neurons and interprets their output through spike rates (i.e. rate coding). Let $T$ denote the number of time steps, with the initial condition $u_i^l[0] = 0$, we can iteratively update the membrane potential using Eq. (3) until $t = T$. Then substitute $z_i^l[t]$ with Eq. (4), and we can write:

$$\frac{\sum_{t=1}^T S_i^l[t]}{T} = \sum_j w_{ij}^l \frac{\sum_{t=1}^T S_j^{l-1}[t]}{T} + \sum_{t=1}^T \frac{b_i^l}{T} - \frac{u_i^l[T]}{T} \tag{5}$$

See Appendix A.1 for a detailed derivation. Note that both sides of the equation are divided by $T$ to better highlight the interpretation of $\sum_{t=1}^T S_i^l[t]/T$ as a "rate". It defines the relationship between neuron's input rate and output rate and can be directly related to the forward pass in a ReLU-activated ANN:

$$a_i^l = \max\left(\sum_j \hat{w}_{ij}^l a_j^{l-1} + \hat{b}_i^l, 0\right) \tag{6}$$

where $a_i^l$ denotes the ANN activation, $\hat{w}_{ij}^l$ and $\hat{b}_i^l$ denote the weight and bias, respectively. Note that in Eq. (5) we have: 1) $\sum S[t]/T > 0$, and 2) $u_i^l[T]/T$ becomes negligible as $T$ increases. These observations suggest that mapping ANN activations to SNN spike rates can be achieved by simply using the scaled ANN weights[2] and bias.

However, with fewer time steps, the spike rate $\sum S_i^l[t]/T$ can only encode a limited number of activations, while the perturbation introduced by $u_i^l[T]/T$ increases. These factors together result in a rapid increase in conversion loss. This issue is inherent in any encoding scheme that relies on quantities linearly related to the time steps. Therefore, our goal is to incorporate nonlinearity into the encoding process to enhance the expressiveness of spike trains.

## 4 METHODS

### 4.1 ASSIGNING WEIGHTS TO SPIKES

We begin by introducing an amplification factor $\beta > 1$ into the soft-reset IF model:

$$u_i^l[t] = \beta u_i^l[t-1] + z_i^l[t] - S_i^l[t] \tag{7}$$

---

[2] The spile amplitude $\theta^l$ is finally normalized to 1 for simplicity of implementation, which is achieved by absorbing it into the synaptic weights. Consequently, the ANN weights still need to be scaled by a certain factor. Note that $\theta^l$ is typically determined based on the number of time steps and the range of ANN activations in layer $l$.

Following the same derivation as in Eq. (5), we can write:

$$u_i^l[T] = \sum_j w_{ij}^l \sum_{t=1}^{T} \beta^{T-t} S_j^{l-1}[t] + \sum_{t=1}^{T} \beta^{T-t} b_i^l - \sum_{t=1}^{T} \beta^{T-t} S_i^l[t] \tag{8}$$

The detailed derivation can be found in Appendix A.1. As expected, the input at time $t_i^{l,f}$ raises the membrane potential by $\theta^l \beta^{T-t_i^{l,f}}$. As shown in Fig. 1, for a sequence of length $T$, this enables the use of $\sum_{t=1}^{T} \beta^{T-t} S_i^l[t]$ rather than $\sum_{t=1}^{T} S_i^l[t]$ to map the ANN activation. Note the spike at time $T$ still encodes $\theta^l$, which is the minimum value a spike can represent and determines the granularity of encoding.

**Definition 1.** Let $v$ denote the target value. The encoding is considered accurate, denoted as $S_i^l[t] \sim v$, as long as $\left| \sum_{t=1}^{T} \beta^{T-t} S_i^l[t] - v \right| < \theta^l$.

We allow a discrepancy of one spike amplitude between the target and encoded value, which can be considered as a quantization error due to the finite number of time steps. According to Eq. (8), our method can theoretically encode the same number of activations as linear encoding method while log-compressing the number of required time steps. Meanwhile, Eq. (8) serves as the core equation for ANN-SNN conversion. We can rewrite it as follows:

$$\sum_{t=1}^{T} \beta^{T-t} S_i^l[t] = \sum_j w_{ij}^l \sum_{t=1}^{T} \beta^{T-t} S_j^{l-1}[t] + \sum_{t=1}^{T} \beta^{T-t} b_i^l - u_i^l[T]$$

By comparing the above equation with Eq. (6), and noting that $\sum_t \beta^{T-t} S_i^l[t] \geq 0$, we can conclude:

**Observation 1.** Let $S_j^{l-1}[t] \sim a_j^{l-1}$, and set $w_{ij}^l = \hat{w}_{ij}^l$ and $b_i^l = \hat{b}_i^l / \sum_t \beta^{T-t}$, respectively. To reduce encoding errors in layer $l$, the residual membrane potential $u_i^l[T]$ should be minimized.

Building on this insight, we now identify the factors influencing $u_i^l[T]$, as formalized in the following theorem:

**Theorem 1.** *Making $u_i^l[T] < \epsilon$ is equivalent to satisfying the following equation:*

$$\forall t_0 \in \{1, 2, \cdots, T\},$$

$$\beta^{T-t_0+1} u_i^l[t_0 - 1] + \sum_j w_{ij}^l \sum_{t=t_0}^{T} \beta^{T-t} S_j^{l-1}[t] + \sum_{t=t_0}^{T} \beta^{T-t} b_i^l < \epsilon + \sum_{t=t_0}^{T} \theta^l \beta^{T-t} \tag{9}$$

The second term on the right-hand side of Eq. (9) represents the maximum value a spike train after $t_0$ can encode. This imposes constraints on both the subsequent input and the membrane potential carried over from the preceding step (the left-hand side of the equation). Theorem 1 provides the mathematical foundation for the next section, with its detailed derivation available in Appendix A.2.

## 4.2 Incorporating Negative Spikes

Rueckauer et al. (2017) reported that large activation values in ANNs are rare, with most values concentrated within a smaller range. This suggests that when mapped to weighted spike trains, the majority of spikes will occur in the later time steps (as these spikes encode smaller values). As a consequence, Eq. (9) becomes difficult to satisfy as $t_0$ approaches $T$: the left-hand side contains a large amount of input, while the right-hand side provides limited encoding capacity. This mismatch ultimately results in an increase in $u_i^l[T]$, leading to a further shift in the spike distribution.

To illustrate this more clearly, we have plotted the spike distribution in the first and last layers of VGG-16 (in red) in Fig. 2, alongside the distribution of the average residual membrane potential across all neurons. As shown, the spike distribution in the last layer shifts significantly toward later time steps compared to the first layer, a phenomenon we refer to as the temporal coupling of weighted spikes. Additionally, the residual membrane potential exhibits a distribution resembling random noise, and our experimental results indicate that this leads to nearly random classification performance. Therefore, a new neuron model is needed to enable effective computation with weighted spikes.

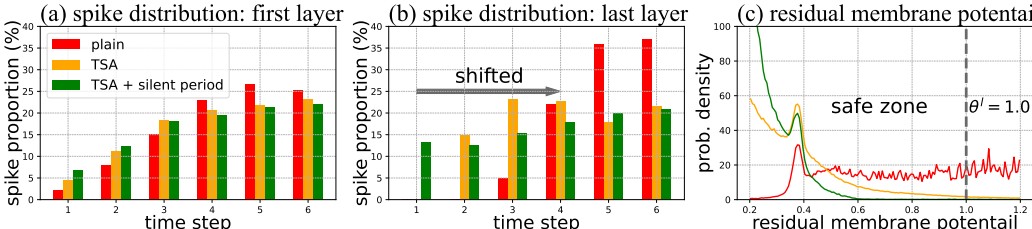

Figure 2: (a) Spike distribution in the first layer of VGG-16. (b) Spike distribution in the last layer of VGG-16. (c) Average residual membrane potential across all neurons in VGG-16. The red data corresponds to the self-amplifying IF neuron model, the orange data corresponds to the TSA model, and the green data further incorporates a one-step silent period.

---

**Algorithm 1:** The forward method of TSA

---

**Input:** input $X$ of shape [BT, C, H, W], length of silent period $L$, spike amplitude $\theta$
**Output:** output spike train $S$ of shape [BT, C, H, W]
reshape and then pad $X$ with zeros to shape [(T+L), B, C, H, W];
membrane potential $M \leftarrow$ zeros_like $(X[0])$, threshold $v \leftarrow \frac{1}{2}\theta\beta^L$;
**for** $0 \leq i \leq L$ **do**
   |   $M \leftarrow \beta M + X[i]$;                                /* silent period */
**end**
**for** $i = 0$ **to** $T - 1$ **do**
   |   $M \leftarrow \beta M + X[i+L]$;                     /* accumulate input */
   |   $S[i] \leftarrow (M \geq v).\text{float}() - (M \leq -v).\text{float}()$ ;    /* fire ternary spikes */
   |   $M \leftarrow M - 2v \times S[i]$;                /* over firing & soft reset */
**end**

---

### 4.2.1 TERNARY SELF-AMPLIFYING NEURON MODEL

Based on the above analysis, our approach begins by encouraging spikes to be generated as early as possible. The key idea is to lower the firing threshold and incorporate negative spikes into the encoding scheme to correct the excess information caused by over-spiking.

We set the positive firing threshold to $\frac{1}{2}\theta^l$ and introduce a negative threshold of $-\frac{1}{2}\theta^l$ into the neuron model, which triggers a negative spike when $o_i^l[t]$ falls below it. Notably, on the left side of Eq. (9), this adjustment not only shifts the input spikes to earlier time steps, but also reduces $u_i^l[t_0 - 1]$. The coefficient $\frac{1}{2}$ is selected to confine both positive and negative membrane potential within a narrow and balanced range. Given the above characteristics, we designate the coding method as the CSS coding scheme and the neuron model as the TSA neuron.

Next, we establish the connection between the ANN and SNN using the proposed methods. Based on Observation 1, we present the following theorem:

**Theorem 2.** *Let $S_j^{l-1}[t] \sim a_j^{l-1}$, $w_{ij}^l = \hat{w}_{ij}^l$, and $b_i^l = \hat{b}_i^l/\sum_t \beta^{T-t}$. Then $S_i^l[t] \sim a_i^l$, provided that $\left| u_i^l[T] \right| < \theta^l$.*

Note $S_i^l[t]$ can now represent negative activations with negative spikes. To handle this, we constrain the absolute value of $u_i^l[T]$ and apply additional logic to zero out sequences that encode negative values (a ReLU counterpart). The above theorem supposes that the input has been encoded and then provides the method for output encoding. Next, we give the method to encode the network input:

**Theorem 3.** *Let the input pixel value be $a_i^0$ and $\beta \leq 2$. By Initializing the membrane potential $u_i^0[0]$ with $a_i^0/\beta^T$, the resulting spike train $S_i^0[t] \sim a_i^0$ with $T$ steps.*

The proofs of the above two theorems can be found in Appendix A.2. By encoding input with Theorem 3 and constraining $u_i^l[T]$ of each hidden layer within the requirements of Theorem 2, an ANN is then converted to a CSS-coded SNN.

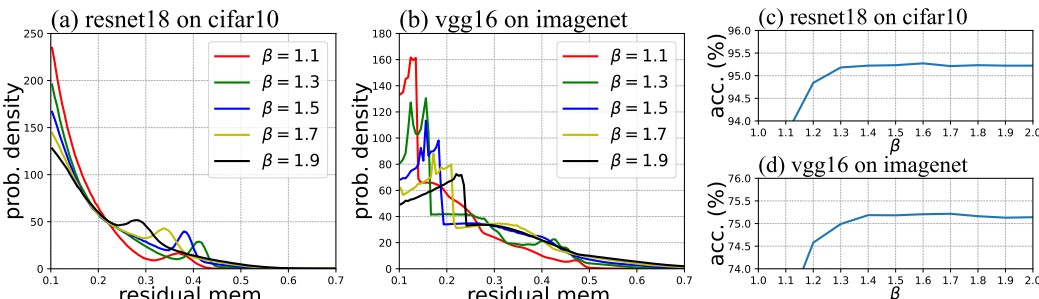

Figure 3: Impact of $\beta$ on residual membrane potential and accuracy. Membrane potentials are normalized by the spike amplitude. (a) and (b) show the residual membrane potential distributions under different $\beta$. (c) and (d) show accuracy variations corresponding to different $\beta$.

### 4.2.2 ONE-STEP SILENT PERIOD

Although the TSA neuron effectively controls $u_i^l[T]$ within an acceptable range, the results in Fig. 2 (orange) demonstrate that temporal coupling persists. Inspired by the layerwise processing manner in TTFS coding (Stanojevic et al., 2022), we incorporate a one-step silent period into the TSA neuron model. During this period, neurons integrate input and perform stepwise weighting but are prohibited from firing. This one-step output delay introduces a new term, $\theta^l \beta^{T-t_0-1}$ (i.e. spike from $t = t_0 - 1$), to the right-hand side of Eq. (9), making it easier to minimize $u_i^l[T]$.

Since the input information is amplified by $\beta$ after the silent period, the firing threshold is adjusted to $\frac{\beta}{2}\theta^l$ accordingly. Similarly, the membrane potential is reduced by $\beta\theta^l$ for reset. In Algorithm 1, we provide pseudo code for the forward propagation process of TSA neurons. The mathematical description of the TSA neuron model can be found in Appendix A.5.

The silent period method assigns distinct computation time windows to TSAs at different depths, aligning with the temporal shift of the input spike distribution. This partially sacrifices synchronous processing at each time step, leading to increased output latency. For an $L$-layer network, the output layer will start to fire spikes only after $L$ steps. However, with each layer operating in a pipelined manner, the efficiency and accuracy gain outweighs the drawback of the increased latency. In Appendix A.4, we set up an ablation study to evaluate the effectiveness of this trade-off.

## 5 EXPERIMENTS

### 5.1 EXPERIMENTAL SETUP

**Determination of Spike Amplitude.** We use a strategy similar to rate coding (Rueckauer et al., 2017) to derive a suitable $\theta^l$: after observing the ANN activations over a portion of the training set, we calculate the 99.99th percentile $p^l$ of the activation distribution, and then set $\theta^l$ to $p^l/\sum_t \beta^{T-t}$.

This setting ensures that the vast majority of the activations remain below the maximum encodable value, $\sum_t \theta^l \beta^{T-t}$, and increases the network's robustness to outlier activations. For ImageNet, we further fine-tune the spike amplitude; additional details on this process can be found in Appendix A.3.

**Choice of the Amplification Factor.** To investigate the impact of $\beta$, we conducted experiments using ResNet-18 on CIFAR-10 and VGG-16 on ImageNet. We plotted the distribution of the residual membrane potential for different values of $\beta$ (which serves as an indicator of temporal coupling) and provided the corresponding accuracy curves in Fig. 3

As shown, accuracy generally increases and then decreases as $\beta$ varies. As $\beta$ increases, the weight difference between spikes grows, causing input spikes to cluster at later time steps and increasing the likelihood of temporal coupling. Conversely, a decrease in $\beta$ requires a larger $\theta^l$ to provide the same encoding capacity, which in turn leads to a higher quantization error. Notably, for $\beta \geq 1.4$, the network exhibits relatively stable performance, with the accuracy difference becoming less pronounced as $\beta$ increases.

Table 2: Coding time steps and inference latency under different neural coding schemes, evaluated on CIFAR-10 and ImageNet datasets.

| | Methods | Architecture | ANN Accuracy | Coding Scheme | Coding Time Step | Inference Latency | SNN Accuracy |
|---|---|---|---|---|---|---|---|
| **CIFAR-10** | FS-conversion (Stöckl & Maass, 2021) | ResNet-20 | 91.58% | FS | 10 | 200 | 91.45% |
| | TTRBR (Meng et al., 2022) | ResNet-18 | 95.27% | rate | 128 | 128 | 95.18% |
| | TSC (Han & Roy, 2020) | VGG-16 | 93.63% | TSC | 512 | 512 | 93.57% |
| | LC-TTFS (Yang et al., 2023) | VGG-16 | 92.79% | TTFS | 50 | 800 | 92.72% |
| | Exact mapping (Stanojevic et al., 2023) | VGG-16 | 93.68% | TTFS | 64 | 1024 | 93.64% |
| | Calibration (Li et al., 2021) | VGG-16 | 95.72% | rate | 128 | 128 | 95.65% |
| | OPI (Bu et al., 2022) | VGG-16 | 94.57% | rate | 128 | 128 | 94.50% |
| | **CSS-SNN** | ResNet-20 | 92.10% | CSS | 12 | 32 | 92.06% |
| | | ResNet-18 | 95.24% | | 12 | 30 | 95.30% |
| | | VGG-16 | 95.89% | | 10 | 26 | 95.88% |
| **ImageNet** | OPI (Bu et al., 2022) | VGG-16 | 74.85% | rate | 256 | 256 | 74.62% |
| | TSC (Han & Roy, 2020) | VGG-16 | 73.49% | TSC | 1024 | 1024 | 73.33% |
| | RMP-SNN Han et al. (2020) | VGG-16 | 73.49% | rate | 2048 | 2048 | 72.78% |
| | Calibration (Li et al., 2021) | VGG-16 | 75.36% | rate | 256 | 256 | 74.23% |
| | TSC (Han & Roy, 2020) | ResNet-34 | 70.64% | TSC | 4096 | 4096 | 69.93% |
| | CalibrationLi et al. (2021) | ResNet-34 | 75.66% | rate | 256 | 256 | 74.61% |
| | FS-conversion (Stöckl & Maass, 2021) | ResNet-50 | 75.22% | FS | 10 | 500 | 75.10% |
| | TTRBR (Meng et al., 2022) | ResNet-50 | 76.02% | rate | 512 | 512 | 75.04% |
| | **CSS-SNN** | VGG-16 | 75.34% | CSS | 12 | 28 | 75.24% |
| | | ResNet-34 | 76.42% | | 14 | 48 | 76.22% |
| | | ResNet-50 | 80.85% | | 16 | 66 | 80.10% |

Based on these observations, we set $\beta$ to 1.5 for two key reasons: it lies near the midpoint of the range, and more importantly, it allows for membrane potential amplification through simple shifting and addition operations, which is energy-efficient and facilitates future hardware implementation.

## 5.2 OVERALL PERFORMANCE

In Table 2, we compared the time steps and inference latency under different coding schemes, which reflect the throughput and latency of the network, respectively. Note all SNNs are converted from full-precision ANNs to ensure a fair comparison. Furthermore, the accuracy of the ANNs utilized in each work is also provided.

**Reduction in Coding Time Steps.** The coding time step refers to the number of time steps required to encode the activations into a spike train. This metric indicates how well the encoding scheme can represent information within a given time frame and reflects the efficiency of the method.

For simpler classification tasks such as CIFAR-10, CSS coding scheme demonstrated nearly loss-less conversion with a significant reduction in the number of required time steps. Compared to linear coding schemes like rate coding, CSS reduces time steps by more than tenfold for both VGG-16 and ResNet-18, while simultaneously reducing the conversion loss. While the FS coding scheme also applied weighted spikes and required fewer time steps for ResNet-20, it experienced greater conversion loss compared to our method. On the more complex ImageNet dataset, the higher precision demands for encoding further highlighted the benefits of spike weighting. For example, Li et al. (2021) reported a conversion error exceeding 1% on ResNet-34 with 256 time steps, whereas our method achieved only 0.2% conversion loss with just 14 time steps. FS-coding achieved smaller conversion loss for ResNet-50 with fewer time steps; however, this came at the cost of a latency eight times greater than that of ours.

**Reduction in Inference Latency.** Inference latency refers to the time elapsed from the beginning of input encoding to the receipt of the classification result, and is also measured in time steps. It indicates how efficiently the encoding scheme transmits information through neural computation across the network layers.

In CSS coding scheme, each layer of TSA neurons incorporates a one-step silent period, making the inference latency equal to the sum of layer counts and coding time steps. In contrast, both TTFS coding and FS coding require each layer to wait for the arrival of all inputs. While this approach facilitates lossless conversion, it completely sacrifices the synchronous processing capability of SNNs, leading to increased output latency (i.e. the product of layer counts and coding time steps).

Table 3: Performance of Fast-SNN, Offset, and CSS on ImageNet after converting 3-bit VGG-16. The results for both Fast-SNN (Hu et al., 2023) and Offset (Hao et al., 2023) are self-implemented using their publicly available repositories, ensuring identical pre-conversion ANN accuracy.

| Methods | T=1 | T=2 | T=3 | T=4 | T=5 | T=6 | T=7 | T=8 |
|---|---|---|---|---|---|---|---|---|
| Offset (rate) | **64.90%** (129 steps) | **70.85%** (130 steps) | 72.06% (131 steps) | 72.53% (132 steps) | 72.79% (133 steps) | 72.92% (134 steps) | 73.05% (135 steps) | 73.01% (136 steps) |
| Fast-SNN (rate) | 3.19% (1 step) | 52.74% (2 steps) | 68.13% (3 steps) | 71.26% (4 steps) | 72.21% (5 steps) | 72.64% (6 steps) | 72.87% (7 steps) | 72.97% (8 steps) |
| **CSS** | 2.87% (17 steps) | 68.53% (18 steps) | **72.81%** (19 steps) | **73.25%** (20 steps) | **73.23%** (21 steps) | **73.24%** (22 steps) | **73.23%** (23 steps) | **73.24%** (24 steps) |

For instance, in the CIFAR-10 classification task, the inference latency reported by Stanojevic et al. (2023) on VGG-16 was about 40 times that of our method. FS coding, as a nonlinear encoding scheme, performs well in both coding steps and conversion loss, but its output latency remains a major weakness; on ResNet-20, its latency exceeded that of CSS by over six times. Rate coding enables synchronous processing, but its inference latency is constrained by the large number of coding time steps. In the ImageNet classification task, for example, rate-coded ResNet-34 had a latency five times greater than our method.

## 5.3 COMBINATION WITH QUANTIZED ANNS

SOTA performance SNNs (Hu et al., 2023; Hao et al., 2023) are typically achieved by converting quantized ANNs, while still utilizing rate coding. To ensure that only the encoding scheme varies, we converted ANNs with identical precision. Experiments were conducted using a 3-bit VGG-16 on ImageNet, with the results presented in Table 3. $T$ represents the number of coding time steps, and the inference latency is given in round brackets. Note we set $\beta$ to 2 for quantized networks as this ensures the encoded values align well with the quantized activations. Additionally, the membrane potential amplification can be achieved using only shifting operations.

Compared to Fast-SNN, our method achieved ANN-level accuracy with fewer time steps by leveraging weighted spikes. The IF neurons in Fast-SNN, which do not require a silent period, naturally exhibit better output latency. In contrast, Offset improved accuracy with minimal time steps by calibrating the initial membrane potential of neurons. However, this comes at the cost of significantly increased output latency, as each layer requires $\rho$ time steps (with $\rho = 8$ for ImageNet, consistent with the original paper) to determine how to calibrate the initial membrane potential. This per-layer latency accumulates across the network, similarly to the silent period in our approach.

Theoretically, the benefits of our method become more pronounced with higher ANN bit precision (as shown in Table 2), as spike weighting reduces the number of time steps exponentially. While rate coding also perform well with reduced time steps, they rely heavily on low-bit quantization. This introduces overhead in training and often sacrifices accuracy. CSS coding scheme provides an alternative approach to achieving low time steps in SNNs without relying on aggressive quantization. Furthermore, our method can be seamlessly combined with quantized ANNs, enabling the development of higher-performance SNNs.

## 5.4 ENERGY CONSUMPTION ANALYSIS

In this section, we estimate the energy consumption of our methods[3], with the results summarized in Table 4. The results show that the CSS-coded SNNs achieved at least a fivefold reduction in energy consumption compared to the original ANN. TTFS (Stanojevic et al., 2023) coding demonstrates extremely low energy consumption due to its theoretical minimum spike count. While our method does not inherently exhibit sparse characteristics, the reduction in coding time steps mitigates this disadvantage. By further compressing the number of time steps, our approach achieved a 10% reduction in energy consumption compared to TTFS. Additionally, we include Fast-SNN (Hu et al., 2023) as a strong baseline for (signed) rate coding. The results show that our method outperformed

---

[3]Energy consumption measurements were performed using the code from `https://github.com/iCGY96/syops-counter`

Table 4: Energy consumption of SNNs on CIFAR-10.

| Methods | Arch. | Accuracy | T | Latency | SyOPs (ACs) | MACs | Energy Consumption |
|---|---|---|---|---|---|---|---|
| ANN | VGG-11 | 93.82% | N/A | N/A | 0 | 153.2M | 0.7047mJ |
| **CSS** | VGG-11 | 93.78% | 8 | 19 | 0 | 132.4M | 0.1191mJ |
| ANN | VGG-16 | 95.88% | N/A | N/A | 0 | 313.88M | 1.4438mJ |
| TTFS* | VGG-16 | 93.53% | 64 | 1024 | 120.53M | 0 | 0.1085mJ |
| **CSS** | VGG-16 | 95.84% | 8 | 24 | 308.35M | 0 | 0.2775mJ |
| **CSS**[†] | VGG-16 | 95.14% | 2 | 18 | 102.19M | 0 | 0.0920mJ |
| ANN | ResNet-18 | 95.25% | N/A | N/A | 0 | 2.22G | 10.21mJ |
| rate (Fast-SNN)[†] | ResNet-18 | 95.42% | 7 | 7 | 1.02G | 12.42M | 0.9751mJ |
| rate (Faset-SNN)[†] | ResNet-18 | 95.23% | 6 | 6 | 878.3M | 10.65M | 0.8395mJ |
| **CSS**[†] | ResNet-18 | 95.31% | 3 | 21 | 730.65M | 1.84M | 0.6660mJ |
| **CSS**[†] | ResNet-18 | 95.24% | 2 | 20 | 489.93M | 1.91M | 0.4497mJ |

[†] The results from converting quantized ANNs.
[*] Stanojevic et al. (2023) reported an average spike rate of 38% per neuron on VGG-16, which we used to calculate the SyOPs and estimate the energy consumption.

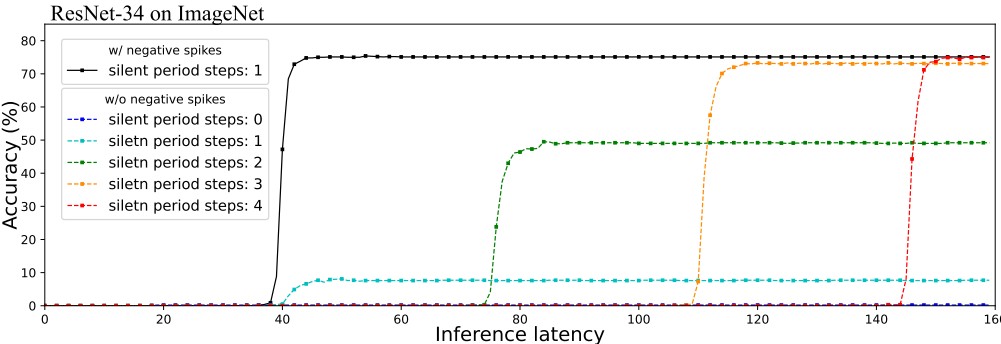

Figure 4: Inference latency with and without negative spikes. The solid line represents the results with negative spikes, while the dashed line indicates the results without negative spikes. The lines of different colors correspond to different lengths of the silent period as shown in the legend.

Fast-SNN with more than a 30% reduction in energy consumption, while maintaining comparable accuracy.

## 5.5 THE ROLE OF NEGTIVE SPIKES

In this section, we validate the role of negative spikes in achieving low-latency nonlinear encoding through an ablation study. Experiments were conducted using ResNet-34 on ImageNet, where we gradually increased the silent period length from zero, in the absence of negative spikes. The results, shown in Fig. 4, reveal that a silent period of at least four steps was required to match the performance gains introduced by negative spikes. This resulted in a nearly 100-step increase in inference latency. The introduction of negative spikes and TSA neurons in the CSS coding scheme is crucial for breaking temporal coupling, distinguishing our approach from other coding schemes that use weighted spikes (Rueckauer & Liu, 2021; Stöckl & Maass, 2021; Kim et al., 2018).

## 6 CONCLUSION AND DISCUSSION

In this work, we compress the coding time steps by assigning weights to spikes, enabling each spike to carry more information. We also introduce negative spikes to break temporal coupling, effectively reducing inference latency. The resulting CSS encoding scheme enhances the throughput, inference speed and energy efficiency of converted SNNs, while minimizing conversion loss.

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

# A APPENDIX

## A.1 PROOFS OF EQUATIONS

**Proof of Eq. (8)** (A similar derivation leads to Eq. (5))

$$u_i^l[T] = \sum_j w_{ij}^l \sum_{t=1}^{T} \beta^{T-t} S_j^{l-1}[t] + \sum_{t=1}^{T} \beta^{T-t} b_i^l - \sum_{t=1}^{T} \beta^{T-t} S_i^l[t] \tag{A1}$$

*Proof.* Starting with the initial condition $u_i^l[0] = 0$ and Eq. (7), we can write:

$$u_i^l[1] = z_i^l[1] - S_i^l[1]$$

Next, we derive the expression for $u_i^l[2]$ by substitute the above into Eq. (7):

$$u_i^l[2] = \beta(z_i^l[1] - S_i^l[1]) + z_i^l[2] - S_i^l[2]$$

We can generalize this process to iteratively compute the membrane potential up to $t = T$:

$$u_i^l[T] = \sum_{t=1}^{T} \beta^{T-t}(z_i^l[t] - S_i^l[t])$$

substituting $z_i^l[t]$ from Eq. (4) and rearranging the terms, we get:

$$u_i^l[T] = \sum_{t=1}^{T} \beta^{T-t}(\sum_j w_{ij}^l S_j^{l-1}[t] + b_i^l - S_i^l[t])$$

Reorganizing the terms by summation yields:

$$u_i^l[T] = \sum_j w_{ij}^l \sum_{t=1}^{T} \beta^{T-t} S_j^{l-1}[t] + \sum_{t=1}^{T} \beta^{T-t} b_i^l - \sum_{t=1}^{T} \beta^{T-t} S_i^l[t]$$

$\square$

## A.2 PROOFS OF THEOREMS

**Theorem 1.** *Making $u_i^l[T] < \epsilon$ is equivalent to satisfying the following equation:*

$$\forall t_0 \in \{1, 2, \cdots, T\},$$

$$\beta^{T-t_0+1} u_i^l[t_0 - 1] + \sum_j w_{ij}^l \sum_{t=t_0}^{T} \beta^{T-t} S_j^{l-1}[t] + \sum_{t=t_0}^{T} \beta^{T-t} b_i^l < \epsilon + \sum_{t=t_0}^{T} \theta^l \beta^{T-t} \tag{A2}$$

*Proof.* We first prove the forward direction. Given that $u_i^l[T] < \epsilon$, we can express it using Eq. (7) and Eq. (4) as follows:

$$\beta u_i^l[T-1] + \sum_j w_{ij}^l S_j^{l-1}[T] + b_i^l < \epsilon + \theta^l \tag{A3}$$

Continue the above process, and we have:

$$\beta^2 u_i^l[T-2] + \sum_j w_{ij}^l \beta S_j^{l-1}[T-1] + \beta b_i^l + \sum_j w_{ij}^l S_j^{l-1}[T] + b_i^l < \epsilon + \theta^l + \beta \theta^l \tag{A4}$$

The above process can be repeated until we obtain an equation involving $u_i^l[0]$. The left-hand side of each equation regarding $u_i^l[t]$, where $t \in \{1, 2, \cdots, T\}$, can be organized to demonstrate that the forward reasoning is valid.

Then we proceed to prove the backward direction. For any $t_0 \in \{1, 2, \cdots, T\}$, by iteratively updating the membrane potential using Eq. (7) from $t = t_0$ until $t = T$, and substitute $z_i^l[t]$ with Eq. (4), we can get:

$$u_i^l[T] = \beta^{T-t_0+1} u_i^l[t_0 - 1] + \sum_j \sum_{t=t_0}^T w_{ij}^l \beta^{T-t} S_j^{l-1}[t] + \sum_{t=t_0}^T \beta^{T-t} b_i^l - \sum_{t=t_0}^T \beta^{T-t} S_i^l[t] \quad \text{(A5)}$$

Note that $\sum_t \beta^{T-t} S_i^l[t] \le \sum_t \theta^l \beta^{T-t}$. Then we can write:

$$u_i^l[T] \le \beta^{T-t_0+1} u_i^l[t_0 - 1] + \sum_j w_{ij}^l \sum_{t=t_0}^T \beta^{T-t} S_j^{l-1}[t] + \sum_{t=t_0}^T \beta^{T-t} b_i^l - \sum_{t=t_0}^T \theta^l \beta^{T-t} \quad \text{(A6)}$$
$$< \epsilon$$

$\square$

**Theorem 2.** *Let $S_j^{l-1}[t] \sim a_j^{l-1}$, $w_{ij}^l = \hat{w}_{ij}^l$, and $b_i^l = \hat{b}_i^l / \sum_t \beta^{T-t}$. Then $S_i^l[t] \sim a_i^l$, provided that $|u_i^l[T]| < \theta^l$.*

*Proof.* Eq. (8) can be organized into the following form:

$$\sum_{t=1}^T \beta^{T-t} S_i^l[t] = \sum_j w_{ij}^l \sum_{t=1}^T \beta^{T-t} S_j^{l-1}[t] + \sum_{t=1}^T \beta^{T-t} b_i^l - u_i^l[T] \quad \text{(A7)}$$

Given that $S_j^{l-1}[t] \sim a_j^{l-1}$, we use $\sigma_j^{l-1}$ to denote the difference between the encoded value and the activation, defined as $\sigma_j^{l-1} = \sum_t \beta^{T-t} S_j^{l-1}[t] - a_j^{l-1}$. Substituting $a_j^{l-1}$ and $\sigma_i^l$ into Eq. (A7), we can write:

$$\begin{aligned}
\sum_{t=1}^T \beta^{T-t} S_i^l[t] &= \sum_j w_{ij}(a_j^{l-1} + \sigma_j^{l-1}) + \sum_{t=1}^T \beta^{T-t} b_i^l - u_i^l[T] \\
&= \sum_j \hat{w}_{ij}(a_j^{l-1} + \sigma_j^{l-1}) + \hat{b}_i^l - u_i^l[T] \\
&= \sum_j \hat{w}_{ij} a_j^{l-1} + \hat{b}_i^l - u_i^l[T] + \sum_j \hat{w}_{ij} \sigma_j^{l-1}
\end{aligned} \quad \text{(A8)}$$

According to Definition 1, we have $-\theta^l < \sigma_j^{l-1} < \theta^l$. Considering that $\theta^l$ is typically kept small to provide fine-grained encoding and $\hat{w}_{ij}$ is generally symmetrically distributed around zero, we can ignore the last term on the right-hand side of the equation. Since $S_i^l[T]$ can encode negative values, we implemented a ReLU counterpart to zero out these spike sequences, corresponding to the $\max(\cdot, 0)$ operation in Eq. (6). Combining Eq. (A8) with the condition $|u_i^l(T)| < \theta^l$ and Definition 1, we can conclude that $S_i^l[t] \sim a_i^l$. $\square$

**Theorem 3.** *Let the input pixel value be $a_i^0$ and $\beta \le 2$. By Initializing the membrane potential $u_i^0[0]$ with $a_i^0 / \beta^T$, the resulting spike train $S_i^0[t] \sim a_i^0$ with $T$ steps.*

*Proof.* We proof this theorem by mathematical induction. Let $\tilde{a}_{i,T}^0$ and $m_T^0$ denote the encoded value and the maximum encodable value, respectively, i.e. $\tilde{a}_{i,T}^0 = \sum_t \beta^{T-t} S_i^0[t]$, $m_T^0 = \sum_t \theta^0 \beta^{T-t}$.

**step 1.** For $T = 1$, it's obvious that:

$$\begin{cases} |\tilde{a}_{i,0}^0 - a_i^0| < \theta^0 & , a_i^0 < m_0^0 \\ \tilde{a}_{i,0}^0 = m_0^0 & , a_i^0 \ge m_0^0 \end{cases} \quad \text{(A9)}$$

**step 2.** Assume the statement is true for $T = t_0$, i.e. we have:

$$\begin{cases} |\tilde{a}_{i,t_0}^0 - a_i^0| < \theta^0 & , a_i^0 < m_{t_0}^0 \\ \tilde{a}_{i,t_0}^0 = m_{t_0}^0 & , a_i^0 \ge m_{t_0}^0 \end{cases} \quad \text{(A10)}$$

Note the relationship between $\theta^0$ and $m_{t_0}^0$: $\theta^0 = \frac{m_{t_0}^0}{1+\beta+\cdots+\beta^{t_0-1}}$. Then we prove that for $T = t_0 + 1$, Eq. (A10) still holds.

**case (1).** For $a_i^0 < m_{t_0+1}^0$: Consider the first $t_0$ steps. It can be observed that this process is equivalent to encoding $a_i^0/\beta$ with $m_{t_0}^0 = \frac{1+\beta+\cdots+\beta^{t_0-1}}{1+\beta+\cdots+\beta^{t_0}}m_{t_0+1}^0$. Then we have:

$$
\begin{cases}
\left| \tilde{a}_{i,t_0}^0 - \dfrac{a_i^0}{\beta} \right| < \dfrac{m_{t_0+1}^0}{1+\beta+\cdots+\beta^{t_0}} & , \ \dfrac{a_i^0}{\beta} < m_{t_0}^0 \\[4mm]
\tilde{a}_{i,t_0}^0 = m_{t_0}^0 & , \ \dfrac{a_i^0}{\beta} \geq m_{t_0}^0
\end{cases}
\tag{A11}
$$

For $\frac{a_i^0}{\beta} < m_{t_0}^0$, we can write:

$$
\beta \left| \tilde{a}_{i,t_0}^0 - \frac{a_i^0}{\beta} \right| < \frac{\beta m_{t_0+1}^0}{1+\beta+\cdots+\beta^{t_0}} < \frac{2m_{t_0+1}^0}{1+\beta+\cdots+\beta^{t_0}} = 2\theta^0
\tag{A12}
$$

For $\frac{a_i^0}{\beta} \geq m_{t_0}^0$:

$$
\begin{aligned}
\beta \left| \tilde{a}_{i,t_0}^0 - \frac{a_i^0}{\beta} \right| &= a_i^0 - \frac{1+\beta+\cdots+\beta^{t_0-1}}{1+\beta+\cdots+\beta^{t_0}}m_{t_0+1}^0 \\
&\leq m_{t_0+1}^0 - m_{t_0+1}^0 \frac{1+\beta+\cdots+\beta^{t_0-1}}{1+\beta+\cdots+\beta^{t_0}} = \theta^0
\end{aligned}
\tag{A13}
$$

According to Theorem 1, $\beta \left| \tilde{a}_{i,t_0}^0 - \frac{a_i^0}{\beta} \right| < 2\theta^0$ is equivalent to $\left| u_i^0[t_0 + 1] \right| < \theta^0$ (as there's neither input nor bias term). Also note $\left| u_i^0[t_0+1] \right| = \left| \tilde{a}_{i,t_0+1}^0 - a_i^0 \right|$. Then we have $\left| \tilde{a}_{i,t_0+1}^0 - a_i^0 \right| < \theta^0$.

**case (2).** For $a_i^0 \geq m_{t_0+1}^0$:

$$
\frac{a_i^0}{\beta} \geq \frac{1}{\beta}m_{t_0+1}^0 > \frac{1+\beta+\cdots+\beta^{t_0-1}}{1+\beta+\cdots+\beta^{t_0}}m_{t_0+1}^0 = m_{t_0}^0
\tag{A14}
$$

Then we have:

$$
\beta \left| \tilde{a}_{i,t_0}^0 - \frac{a_i^0}{\beta} \right| = a_i^0 - \beta\frac{1+\beta+\cdots+\beta^{t_0-1}}{1+\beta+\cdots+\beta^{t_0}}m_{t_0+1}^0 \geq \frac{m_{t_0+1}^0}{1+\beta+\cdots+\beta^{t_0}} = \theta^0
\tag{A15}
$$

which means the neuron will fire a spike at $t = t_0 + 1$, leading to $\tilde{a}_{i,t_0+1}^0 = m_{t_0+1}^0$. Combining case (1) and (2), and we have:

$$
\begin{cases}
\left| \tilde{a}_{i,t_0+1}^0 - a_i^0 \right| < \theta^0 & , \ a_i^0 < m_{t_0+1}^0 \\
\tilde{a}_{i,t_0+1}^0 = m_{t_0+1}^0 & , \ a_i^0 \geq m_{t_0+1}^0
\end{cases}
\tag{A16}
$$

**step 3.** By the principle of mathematical induction, $\forall T \in \mathbb{N}^\star$:

$$
\begin{cases}
\left| \tilde{a}_{i,T}^0 - a_i^0 \right| < \theta^0 & , \ a_i^0 < m_T^0 \\
\tilde{a}_{i,T}^0 = m_T^0 & , \ a_i^0 \geq m_T^0
\end{cases}
\tag{A17}
$$

Considering our initialization strategy for $\tilde{\theta}^0$ and subsequent data-based amplification, we can always ensure that $a_i^0 < m_T^0$. Thus, $\left| \tilde{a}_{i,T}^0 - a_i^0 \right| < \theta^0$, which means $S_i^0[t] \sim a_i^0$. $\square$

### A.3 Spike Amplitude Adjustment

Eq. (9) suggests that increasing $\theta^l$ can relax the constraints on the input. Let $\tilde{\theta}^l$ denote the initial spike amplitude. After getting the initialized value, we use a subset of the training set to perform forward propagation for the CSS-based SNN, and then calculate the 99.9th percentile $u^l$ of the distribution of $u_i^l[T]$ for each layer $l$. If $u^l$ exceeds $\theta^l$, we amplify $\theta^l$ by a factor $s^l$. Note that increasing $\theta^l$ raises the firing threshold at the same time, making the change in $u_i^l[T]$ a complex nonlinear process. To determine a suitable $s^l$, we simplify the problem by assuming that $u_i^l[T]$

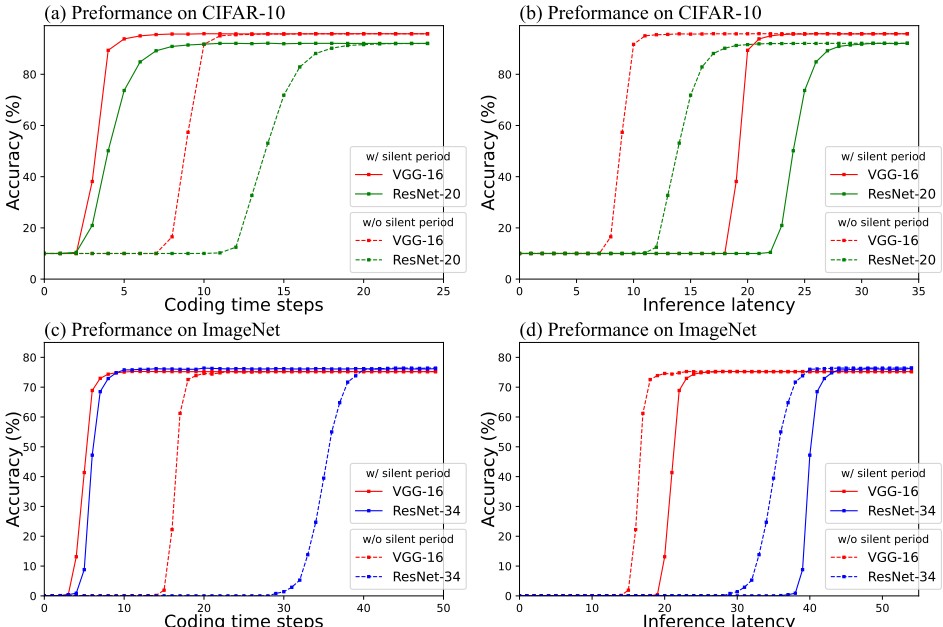

Figure 5: Trade-off between the coding time steps and inference latency. The dashed line represents the results obtained without the silent period, while the solid line represents the results achieved after incorporating the silent period. (a) Coding time steps on CIFAR-10. (b) Inference latency on CIFAR-10. (c) Coding time steps on ImageNet. (d) Inference latency on ImageNet.

accumulates uniformly over time. Thus, if the increment of $\theta^l$ is $\Delta\theta^l$, then $u_i^l[T]$ will decrease by $\Delta\theta^l \sum_t \beta^{T-t}$. Accordingly, $s^l$ is determined using the following equation:

$$
s^l = 1 + \begin{cases} \dfrac{u^l - \theta^l}{\sum_t \beta^{T-t}} & , u^l - \theta^l > 0.05 \cdot \sum_t \beta^{T-t} \\[3mm] \dfrac{u^l - \theta^l - 0.01 \cdot \sum_t \beta^{T-t}}{\sum_t \beta^{T-t}} + 0.04 & , 0.05 \cdot \sum_t \beta^{T-t} \geq u^l - \theta^l > 0.01 \cdot \sum_t \beta^{T-t} \\[3mm] \dfrac{4 \cdot (u^l - \theta^l)}{\sum_t \beta^{T-t}} & , 0.01 \cdot \sum_t \beta^{T-t} \geq u^l - \theta^l > 0 \end{cases}
$$

$$\text{(A18)}$$

To ensure that the spike amplitude can still be effectively adjusted when $u^l$ slightly exceeds $\theta^l$, we increase the value of $s^l$ for this range.

Note that an increase in $\theta^l$ makes the decoupling conditions for layer $l+1$ harder to meet[4]. Consequently, in deeper layers, the initial spike amplitude must be amplified by a large factor. This requires a sufficiently small $\tilde{\theta}^l$ to preserve adequate encoding granularity after scaling, which in turn necessitates a larger number of time steps. We address this issue by delaying the TSA output, which eliminates the need for $\tilde{\theta}^l$ amplification.

It is important to note that this adjustment lacks strict mathematical support and serves as a heuristic for fine-tuning the spike amplitude. The introduction of negative spikes remains the core mechanism for breaking temporal coupling.

## A.4 CODING TIME STEPS VS. INFERENCE LATENCY

According to the analysis in Appendix A.3, relying solely on $\theta^l$ amplification to break temporal coupling would require smaller $\tilde{\theta}^l$ in deeper layers, which leads to an increase in coding time steps. To address this, we introduce a one-step silent period to achieve a trade-off between coding time steps

---

[4]In Eq. (9), the spike amplitude of the previous layer is included in $S_j^{l-1}[t]$.

and inference latency. In this section, we conducted an ablation study to assess the effectiveness of this approach. We performed classification tasks on CIFAR-10 using VGG-16 and ResNet-20, and on ImageNet using VGG-16 and ResNet-34. Fig. 5 (a) and (c) present the relationship between coding time steps and accuracy, while Fig. 5 (b) and (d) show the relationship between inference latency and accuracy.

The experimental results indicate that even with no silent period, deeper networks experience larger latency due to increased coding time steps. This can also be understood as neurons in each layer require time to accumulate membrane potential before firing. Thus, incorporating a silent period has a limited effect on increasing inference latency, but plays a significant role in reducing coding time steps. For example, in ResNet-20 on CIFAR-10, the silent period increased latency from 20 to 30 steps but halved the coding time steps, greatly improving throughput. This effect becomes more pronounced with increased network depth or dataset scale. For instance, with ResNet-34 on ImageNet, the silent period added only about 5 steps to inference latency while reducing coding time steps by approximately 30 steps. Overall, incorporating the silent period effectively reduces the required number of time steps for encoding, substantially improving throughput with minimal impact on latency.

It is important to note that although the requirement ofTheorem 2 can be satisfied by amplitude adjustment on training set, this does not ensure optimal performance on the test set. By contrast, silent period provides a data-independent approach to break temporal coupling, resulting in more stable and consistent performance improvements.

## A.5 MATHEMATICAL DESCRIPTION OF THE TSA NEURON

To generalize the representation, let $T_s$ denote the length of the silent period. For a TSA neuron in the $l$-th layer, when the time step $t \in \{1, 2, \ldots, T_s l\}$, the membrane potential remains $u_i^l[t] = u_i^l[0] = 0$, and no spikes are generated. After this period, the neuron processes inputs in cycles of length $T + T_s$. Let $T_a^k$ denote the start of the $k$-th cycle, i.e. $T_a^k = T_s l + k(T_s + T) + 1$, and let $T_b^k$ denote the end of the cycle, i.e. $T_b^k = T_s l + (k + 1)(T_s + T)$, with $k$ being a natural number. Without loss of generality, we consider the case where $t \in \{T_a^k, T_a^k + 1, \ldots, T_b^k\}$.

The set of its spike times can be expressed as follows:

$$\mathbb{F}_i^l = \left\{ t_i^{l,f} \,\middle|\, \left|o_i^l[t_i^{l,f}]\right| \geq \frac{\theta^l \beta^{T_s}}{2}, \; t_i^{l,f} \in \{T_a^k + T_s, T_a^k + T_s + 1, \cdots, T_b^k\} \right\} \quad (A19)$$

The spike sequence it emits, $S_i^l[t]$, can then be written as:

$$S_i^l[t] = \sum_{t_i^{l,f} \in \mathbb{F}_i^l} \mathrm{sgn}\left(o_i^l[t_i^{l,f}]\right) \theta^l \delta[t - t_i^{l,f}] \quad (A20)$$

where $\delta[\cdot]$ denotes an unit impulse, $\mathrm{sgn}(\cdot)$ is the sign function and $\theta^l$ is the spike amplitude. The update process of the membrane potential can be expressed as follows:

$$u_i^l[t] = \beta u_i^l[t - 1] + z_i^l[t] - \beta^{T_s} S_i^l[t] \quad (A21)$$

where $z_i^l[t]$ denotes the integrated inputs:

$$z_i^l[t] = \sum_j w_{ij}^l S_j^{l-1}[t] + b_i^l \quad (A22)$$

## A.6 PSEUDO CODE FOR CONVERSION PROCESS

See Algorithm 2. The spike amplitude for each layer is determined using the method outlined in Section 4.2.1. Note that this value is absorbed into the weights and bias, so after conversion, the spike amplitudes for TSA neurons are all normalized to 1. For data-based spike amplitude adjustment, we first complete the above conversion process using the initial spike amplitudes. Then, based on the residual membrane potentials observed in the resulting SNN, we update the spike amplitudes. These updated values are then used to re-convert the ANN. The process can be repeated as may times as needed.

---

**Algorithm 2:** Algorithm for ANN-SNN conversion

---

**Input:** ANN model $f_A(\hat{\boldsymbol{W}}, \hat{\boldsymbol{b}})$, encode time steps $T$, amplification factor $\beta$, spike amplitude $\theta^l$
     for each layer $l$, total layer number $L$
**Output:** SNN models $f_S(\boldsymbol{W}, \boldsymbol{b})$
reshape and pad $X$ to [T, B, C, H, W] with zeros to shape [(T+L), B, C, H, W];
membrane potential $M \leftarrow$ zeros_like $(X[0])$, threshold $v \leftarrow \frac{1}{2}\theta\beta^L$;
set CSS encoder for the input layer;                         `/* see Theorem 3 */`
**for** $1 \leq l \leq L$ **do**
   $W^l \leftarrow \frac{\theta^{l-1}}{\theta^l}\hat{W}^l$;                              `/* norms `$\theta^l$` in SNN to 1 */`
   $b^l \leftarrow \frac{1}{\sum_{t=0}^{T-1}\beta^t\theta^l}\hat{b}^l$;
   replace ReLU activation with TSA and ReLU counterpart.
**end**

---

## A.7   IMPLEMENTATION OF THE RELU COUNTERPART

In the actual implementation, we fuse the ReLU counterpart into the TSA neuron model to speed up program execution. We refer to this model as TSA-ReLU neuron. Below, we continue the notation from Appendix A.5 to present the mathematical model of TSA-ReLU. Without loss of generality, we consider the case where $t \in \{T_a^k, T_a^k + 1, \ldots, T_b^k\}$. We use $h_i^l[t]$ and $g_i^l[t]$ to represent the accumulated input and output of TSA-ReLU, respectively:

$$h_i^l[t] = \sum_j w_{ij}^l \sum_{\tau=T_a^k}^{t} \beta^{t-\tau} S_j^{l-1}[\tau] + \sum_{\tau=T_a^k}^{\max(t, T_a^k+T)} \beta^{t-\tau} b_i^l$$

$$g_i^l[t] = \sum_{\tau=\min(T_a^k+T_s, t)}^{t} \beta^{t-\tau} S_i^l[\tau] \tag{A23}$$

Then we set $o_i^l[t]$ according to the following equation:

$$o_i^l[t] = \begin{cases} h_i^l[t] - g_i^l[t] & , h_i^l[t] \geq 0, t \geq T_a^k + T_s \\ -g_i^l[t] & , h_i^l[t] < 0, t \geq T_a^k + T_s \\ 0 & , t < T_a^k + T_s \end{cases} \tag{A24}$$

The firing condition of TSA-ReLU is the same as that of TSA, and is given by Eqs. (A19) and (A20).

