# SUPPLEMENTARY MATERIALS

## 1 COMPARISON WITH STATE-OF-THE-ART METHODS

Table 1: Performance of Fast-SNN, Offset, and CSS on ImageNet after converting 3-bit VGG-16

| Methods | T=1 | T=2 | T=3 | T=4 |
|---|---|---|---|---|
| Offset (rate) | 64.90% (129 steps) | 70.85% (130 steps) | 72.06% (131 steps) | 72.53% (132 steps) |
| Fast-SNN (rate) | 3.19% (1 steps) | 52.74% (2 steps) | 68.13% (3 steps) | 71.26% (4 steps) |
| **CSS** | 2.87% (17 steps) | 68.53% (18 steps) | 72.81% (19 steps) | 73.25% (20 steps) |
| Methods | T=5 | T=6 | T=7 | T=8 |
| Offset (rate) | 72.79% (133 steps) | 72.92% (134 steps) | 73.05% (1235 steps) | 73.01% (136 steps) |
| Fast-SNN (rate) | 72.21% (5 steps) | 72.64% (6 steps) | 72.87% (7 steps) | 72.97% (8 steps) |
| **CSS** | 73.23% (21 steps) | 73.24% (22 steps) | 73.23% (23 steps) | 73.24% (24 steps) |

Table 2: Performance of Fast-SNN, Offset, and CSS on ImageNet after converting 2-bit VGG-16

| Methods | T=1 | T=2 | T=3 | T=4 |
|---|---|---|---|---|
| Offset (rate) | 68.44% (65 steps) | 71.71% (66 steps) | 72.39% (67 steps) | 72.54% (68 steps) |
| Fast-SNN (rate) | 23.82% (1 steps) | 66.98% (2 steps) | 70.99% (3 steps) | 71.92% (4 steps) |
| **CSS** | 19.49% (17 steps) | 71.22% (18 steps) | 72.31% (19 steps) | 72.56% (20 steps) |

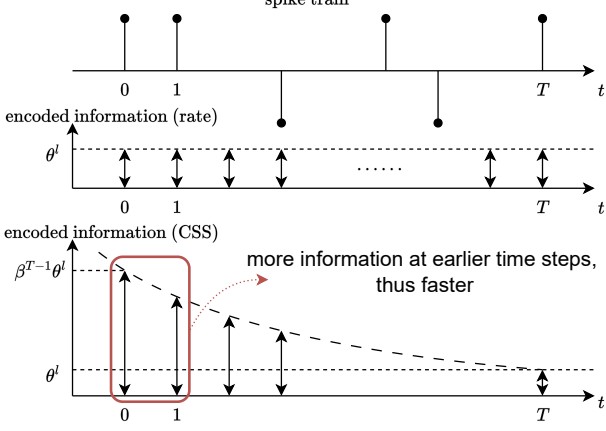

Figure 1: CSS coding assigns more weight to earlier spikes, allowing it to achieve higher performance than rate coding while using fewer time steps.

## 2 THE ROLE OF NEGATIVE SPIKES

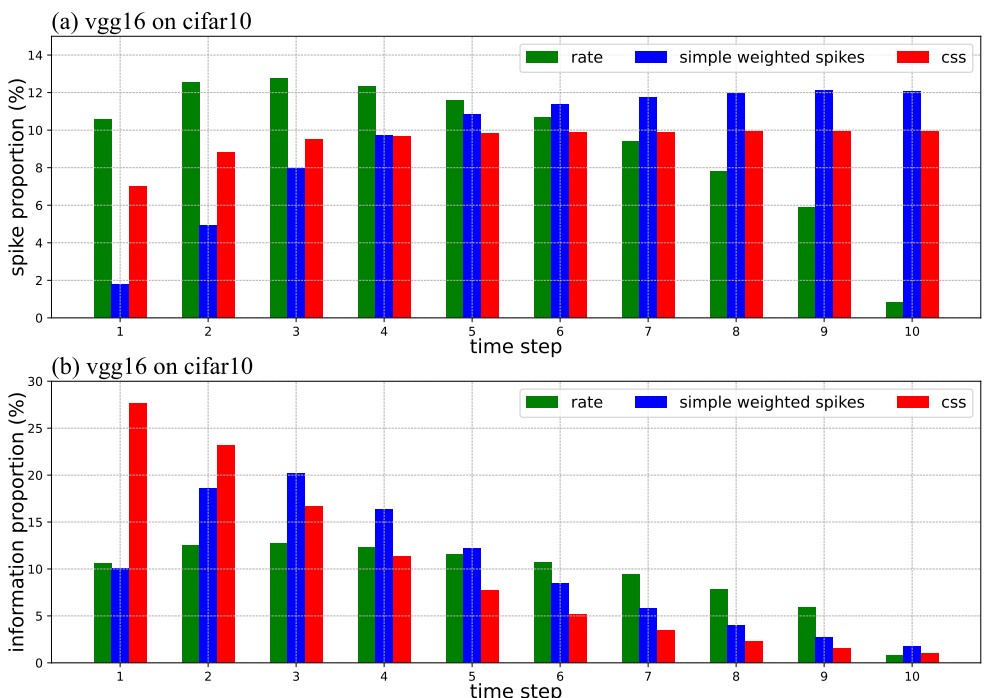

Figure 2: (a) The distribution of spikes across different time steps. Green represents rate coding. Blue represents the case after spike weighting, where the spikes are concentrated in the later time steps. Red represents the case in CSS coding, where we successfully shifted the distribution of weighted spikes toward earlier time steps. (b) The distribution of fired information across different time steps.

Table 3: Changes in accuracy before and after introducing negative spikes. Experiments were performed using VGG-16 on CIFAR-10.

| Coding Scheme | Negative Spikes | T=2 | T=4 | T=6 | T=8 |
|---|---|---|---|---|---|
| rate | × | 10.00% | 47.48% | 94.25% | 95.26% |
| **CSS** | × | 90.93% | 92.94% | 93.12% | 93.22% |
| rate | ✓ | 10.21% | 47.88% | 94.42% | 95.34% |
| **CSS** | ✓ | 95.14% | 95.55% | 95.60% | 95.58% |

# 3 ENERGY CONSUMPTION ANALYSIS

Table 4: Energy consumption of VGG-structured SNNs on CIFAR-10

| Methods | Arch. | Accuracy | T | Latency | SyOP (ACs) | MACs | Energy Consumption |
|---------|-------|----------|---|---------|------------|------|--------------------|
| ANN | VGG-11 | 93.82% | N/A | N/A | 0 | 153.2M | 0.7047mJ |
| **CSS** | VGG-11 | 93.78% | 8 | 19 | 0 | 132.4M | 0.1191mJ |
| ANN | VGG-16 | 95.88% | N/A | N/A | 0 | 313.88M | 1.4438mJ |
| TTFS | VGG-16 | 93.53% | 64 | 1024 | 120.53M | 0 | 0.1085mJ |
| **CSS** | VGG-16 | 95.84% | 8 | 24 | 308.35M | 0 | 0.2775mJ |
| **CSS** | VGG-16 | 95.14% | 2 | 18 | 102.19M | 0 | 0.0920mJ |

Table 5: Energy consumption of Fast-SNN and CSS on CIFAR10 after converting 3-bit ResNet-18.

| Methods | Accuracy | T | Latency | SyOP (ACs) | MACs | Energy Consumption |
|---------|----------|---|---------|------------|------|--------------------|
| ANN | 95.25% | N/A | N/A | 0 | 2.22G | 10.21mJ |
| Fast-SNN (rate) | 95.42% | 7 | 7 | 1.02G | 12.42M | 0.9751mJ |
| Fast-SNN (rate) | 95.23% | 6 | 6 | 878.3M | 10.65M | 0.8395mJ |
| **CSS** | 95.31% | 3 | 21 | 730.65M | 1.84M | 0.6660mJ |
| **CSS** | 95.24% | 2 | 20 | 489.93M | 1.91M | 0.4497mJ |

# 4 SELECTION OF THE AMPLIFICATION COEFFICIENT

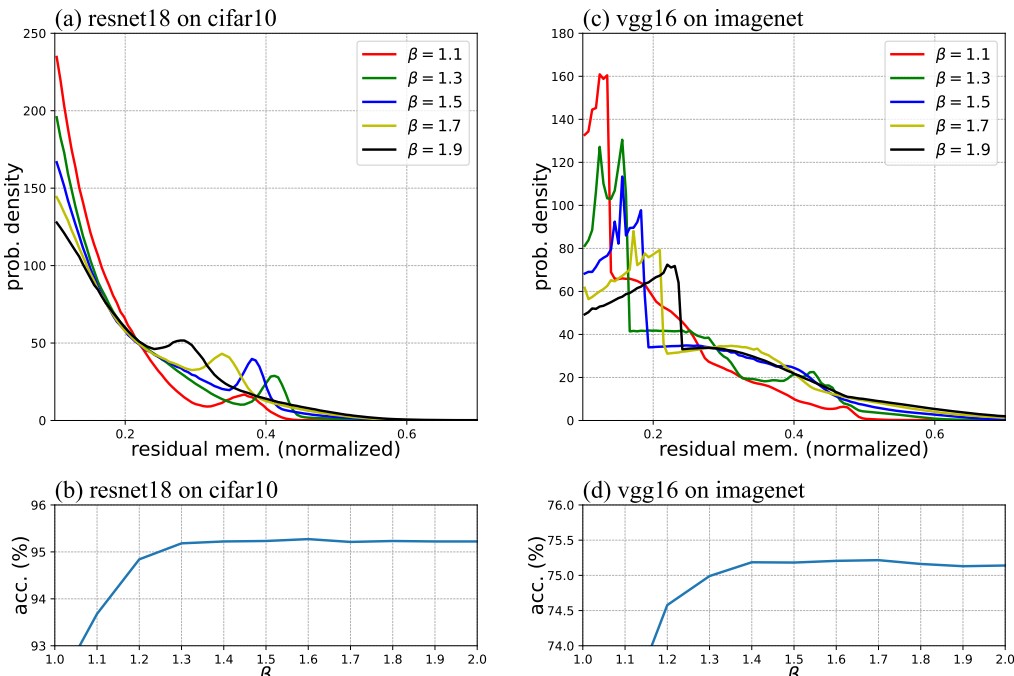

Figure 3: Impact of amplification coefficient on residual membrane potential and accuracy. All the membrane potentials are normalized. Note the firing threshold is 0.5. For ResNet-18 on CIFAR-10: (a) Residual membrane potential distributions under different $\beta$. (b) Accuracy variations corresponding to different $\beta$. For VGG-16 on ImageNet: (c) Residual membrane potential distributions under different $\beta$. (d) Accuracy variations corresponding to different $\beta$.