# OpenReview forum: "Canonic Signed Spike Coding for Efficient Spiking Neural Networks"
_ICLR.cc/2025/Conference — Submitted to ICLR 2025_

### Official Review · Reviewer_DPaT · 2024-10-27

**Soundness:** 1
**Presentation:** 2
**Contribution:** 1
**Rating:** 3
**Confidence:** 4

**Summary:**

This paper proposes a new ANN-SNN Conversion learning framework, which is based on Canonic Signed Spike (CSS) coding scheme and Ternary Self-Amplifying (TSA) neuron model.

**Strengths:**

1. The authors conduct theoretical analysis and proof for the ANN-SNN Conversion process based on CSS coding scheme.

**Weaknesses:**

1. The concept of firing negative spikes has already been proposed by [1].
2. This paper involves firing negative spikes, however, most of the comparative works selected by the authors in Table 2 are based on firing binary spikes. Furthermore, training high-performance converted SNN models under low time latency is currently one of the research focuses in the domain of ANN-SNN Conversion, but the authors do not reveal the performance of CSS under low time latency ($\leq 4$ time-steps) as demonstrated in [1, 2, 3].

[1] Li C, et al. Quantization framework for fast spiking neural networks. Frontiers in Neuroscience, 2022.

[2] Bu T, et al. Optimal ANN-SNN conversion for high-accuracy and ultra-low-latency spiking neural networks. ICLR, 2022.

[3] Hao Z, et al. Bridging the gap between ANNs and SNNs by calibrating offset spikes. ICLR, 2023.

========================

After reading the authors' further response, I will provide the following points of views for AC and other Reviewers to consider:

* The so-called "silent period" and "fire ternary spikes", as shown in Algorithm 1, can it be effectively implemented on neuromorphic hardware without causing additional time latency and energy consumption?
* Is it fair to directly compare CSS with other vanilla conversion methods, which are based on firing binary spikes and do not introduce special mechanisms (e.g. silent period) ?
* In Tab.3, COS [3] is a conversion method that emits spikes layer by layer, rather than a method that emits spikes step by step. But the authors directly calculate the time latency based on "$T+\rho\times(\textbf{Network Layers})$" time-steps, without respecting the $T+\rho$ statistical standard used in the original paper. Is this reasonable?

In their response, the authors think that my rating opinion is vague, so the above three points are the relatively detailed reasons I further provided.

**Questions:**

See Weakness Section.

---

> ### Author Response · Authors · 2024-11-25
>
> Thank you for your feedback. We have conducted comprehensive supplementary experiments and will do our best to address your concerns.
>
> * The concept of firing negative spikes has already been proposed by [1].
>
> Indeed, we are not the first to introduce negative spikes in SNNs. However, the role that negtive spikes play is very different between the two papers. The primary contribution of [1] lies in converting quantized ANNs, which enables rate coding schemes to function effectively. In their work, negative spikes are used to allow neurons to respond to negative inputs, as described in their paper:
>
> > A negative spike will be generated when the membrane potential is smaller than zero and the total spike count generated by this neuron is greater than zero.
>
> In contrast, our contribution focuses on proposing a novel encoding scheme. Due to spike weighting, we observe that spikes tend to concentrate in later time steps. To address this, we reduce the threshold and introduce negative spikes to counteract encoding errors caused by early firing. This approach primarily aims to adjust the spike distribution and resolve the issue of temporal coupling.
> To better illustrate this, we conducted comparative experiments, the results of which are included in the public comment section. Our findings indicate that under rate coding, negative spikes primarily respond to negative inputs and are not critical for performance improvement. However, in our encoding scheme, negative spikes play a pivotal role in correcting excessive firing and are essential for final accuracy.
> To the best of our knowledge, we are the first to incorporate negative spikes into a nonlinear encoding scheme, which breaks temporal coupling and significantly reduces output latency.
>
> * This paper involves firing negative spikes, however, most of the comparative works selected by the authors in Table 2 are based on firing binary spikes. Furthermore, training high-performance converted SNN models under low time latency is currently one of the research focuses in the domain of ANN-SNN Conversion, but the authors do not reveal the performance of CSS under low time latency (time-steps) as demonstrated in [1, 2, 3].
>
> Thank you for your suggestions. As mentioned in the Introduction, recent works (e.g., [1, 2, 3]) have successfully reduced the number of time steps required for encoding by converting quantized ANNs. This reduction is largely attributed to the significantly smaller number of activation values in quantized networks. While effective, this approach addresses a different aspect of the conversion process and operates independently from our method. The primary contribution of this paper lies in introducing a novel encoding method that does not rely on quantization.
>
> We have included comparisons with state-of-the-art methods (Fast-SNN and [3]) in our public comment. All methods were tested by converting the same low-precision ANN. From the experimental results, our method achieves optimal performance with fewer time steps compared to Fast-SNN, underscoring the inherent advantage of spike weighting. Offset performs relatively well at 1 or 2 time steps but at the cost of significantly increased latency. Furthermore, our method surpasses its accuracy when the number of time steps increases to around 3. We encourage you to refer to our public comment for more detailed information.
>
> From a theoretical perspective, our encoding scheme is expected to outperform rate coding because we enhance the encoding capacity of spikes by weighting them. We have visually illustrated the differences between the two encoding schemes in Fig 1. of the supplementary material, and we hope you find it helpful for reference.

---

> ### Author Response · Authors · 2024-11-27
>
> __Dear Reviewer DPaT__,
>
> We hope this message finds you well.
>
> We would like to kindly follow up on the feedback regarding our rebuttal submission for the manuscript titled “Canonic Signed Spike Coding for Efficient Spiking Neural Networks.” We truly appreciate the thorough review and valuable comments you have provided thus far.
>
> Understanding that the review process can be time-consuming, we would be grateful if you could share an update on the current status of our rebuttal. If there is anything further we can do or clarify, please don’t hesitate to let us know.
>
> We sincerely appreciate your time and effort in reviewing our work and look forward to your feedback.
>
> __Best regards__,
>
> __Authors of Submission 10929__

---

> ### Comment · Reviewer_DPaT · 2024-11-28
>
> Thanks for the authors' response. I have comprehensively considered the contributions of this work in terms of innovation and experimental performance, but I ultimately tend to believe that this work has not yet reached the acceptance criteria of ICLR. Therefore, I will tend to maintain my rating.

---

> > ### Author Response · Authors · 2024-11-29
> >
> > Thank you for your comment. Based on the questions you have raised, we believe we have addressed them effectively. We propose a novel encoding method that goes beyond rate coding and temporal coding. Our experimental results show that CSS-coded SNNs can achieve higher efficiency and accuracy. Additionally, our method is hardware-friendly, providing a foundation for the practical implementation of this approach. We believe our method has the potential to become a primary encoding scheme for future SNN applications.
> >
> > We would like to defend the novelty of the negative spike in our work once again. The contribution of our paper lies in the introduction of a new encoding scheme, with one of the core ideas being compressing the time steps through spike weighting. This approach directly addresses the issue of excessive time steps in both rate coding and temporal coding, without relying on aggressive quantization. Therefore, we do not believe that the introduction of negative spikes in [1] impacts the novelty of our approach, as they still use rate coding. The use of negative spikes in rate coding does not imply that negative spikes should be incorporated into other encoding schemes, as they address fundamentally different problems. We have experimentally verified this point, and we kindly refer you to the section "The Role of Negative Spikes" in the public comment for further details.
> >
> > Regarding the choice of comparison methods, we have clarified this in the public comment and have supplemented the experiments accordingly. A fair comparison requires the original ANN to be of consistent bit precision. To address your concerns, we have added comparisons with Offset (i.e., [3]), Fast-SNN (both SOTA SNNs), along with energy analysis. From both theoretical and experimental perspectives, our method offers advantages. Furthermore, by directly converting a full-precision ANN, our method even outperforms certain rate-coded methods that convert quantized ANNs (e.g., [2]), in both coding time steps and output latency. This further highlights the efficiency of our approach, especially when considering the overhead or accuracy loss involved in training QCFS-ANNs or 3/2-bit ANNs.
> >
> > We are encouraged that other reviewers have recognized the novelty of our approach, and our additional experiments have addressed their concerns regarding performance. We believe it is necessary for you to specify the concrete issues with the paper, rather than dismissing it with a vague statement like "this work has not yet reached the acceptance criteria of ICLR." Such general comments could apply to any paper.
> >
> > As ICLR submissions and reviews are public, we expect that the relevant researchers will assess whether our paper demonstrates sufficient innovation and contribution, and whether your review is reasonable and convincing.

---

> ### Author Response · Authors · 2024-12-02
>
> Thank you for your valuable questions. We apologize for the delayed response. Below, we will address your concerns to the best of our ability.
>
> * The so-called "silent period" and "fire ternary spikes", as shown in Algorithm 1, can it be effectively implemented on neuromorphic hardware without causing additional time latency and energy consumption?
>
> Thank you for your thoughtful question. We would like to clarify that _our approach is indeed quite hardware-friendly_. Compared to traditional rate coding with IF neurons, our method requires  additional logic, which includes: __multiplying the membrane potential by $\beta$, controlling the silent period, and handling input and output of negative spikes__.
>
> __The control signal for the silent period__ can be efficiently generated by a state register. Specifically, when a neuron starts its computation, the register outputs a value of 0, and one clock cycle later, it stays at 1 until the computation is reset. This control signal is shared across many neurons, resulting in minimal hardware overhead.
>
> __For the membrane potential amplification__, setting $\beta=2$ allows us to achieve the desired effect through a simple shift operation. We use shift registers to perform the operation, which also double as the registers storing the membrane potential. A MUX is added to select between the shift operation and accumulation.
>
> __handling input negative spikes__ is straightforward and only requires a two's complement addition, which can be efficiently performed by the adder. __For negative spike emission__, we use the most significant bit (MSB) of the membrane potential to determine its polarity. If the MSB is 1, we take the two’s complement and compare it with the positive threshold in the comparator. This approach effectively compares the absolute value of the membrane potential with the threshold, requiring only a single comparator, which incurs minimal hardware overhead. Based on the comparison and the MSB value, we can determine both the presence and the polarity of the spike.
>
> Thus, __the added hardware mainly consists of a set of multiplexers and a two's complement unit__. The silent period control module and the spike emission determination module contribute minimal hardware overhead, which is negligible. __The shift operation contributes most to the increased energy consumption__, and we have accounted for it in the *Energy Consumption Analysis* section. Specifically, we estimate the energy consumption of the shift operation to be roughly equivalent to a SyOP (AC), which is an overestimate.
>
> Based on our experimental results and the analysis above, overall, _our method is hardware-friendly and does not have a large impact on latency or energy consumption_.
>
>
> * Is it fair to directly compare CSS with other vanilla conversion methods, which are based on firing binary spikes and do not introduce special mechanisms (e.g. silent period) ?
>
> Thank you for your insightful question. Our core contribution lies in a novel encoding scheme, where negative spikes are an integral part of the method. Therefore, we believe that our practice is reasonable and effectively highlights the novelty of the proposed method, particularly when compared to other non-linear coding schemes.
>
> In fact, __mechanisms analogous to the silent period exist in other coding schemes__ as well. For example, TTFS coding and FS coding both employ a layer-by-layer processing approach, which can be interpreted as setting $L$ (the length of the silent period) equal to $T$ (the coding time steps). Additionally, __negative spikes are also utilized__ in Fast-SNN (rate coding) and TSC (temporal switch coding), which are included as comparison methods in our study.

---

> > ### Author Response · Authors · 2024-12-02
> >
> > * In Tab.3, COS [3] is a conversion method that emits spikes layer by layer, rather than a method that emits spikes step by step. But the authors directly calculate the time latency based on $T+\rho\times N_{\mathit{layers}}$ time-steps, without respecting the $T+\rho$ statistical standard used in the original paper. Is this reasonable?
> >
> > Thank you for your thoughtful question. To compare different encoding schemes, we use two key metrics: coding time steps and inference latency, which together provide a reasonable reflection of the network's performance in terms of throughput and latency. We have defined both metrics in the experimental section of the paper as follows:
> >
> > > - **Coding time step** refers to the number of time steps required to encode the activations into a spike train.
> > > - **Inference latency** refers to the time elapsed from the start of input encoding to the receipt of the classification result, and is also measured in time steps.
> >
> > For COS (rate coding), each layer’s neurons require $\rho$ time steps to determine how to adjust the initial membrane potential before spikes are sent to the next layer. Following the above definitions, we calculate the output latency for COS as $T+\rho\times N_{\mathit{layers}}$, with $T$ being the coding time steps. Note that the total latency is not $(T+\rho)\times N_{\mathit{layers}}$, as each layer can process input in a pipelined manner.
> >
> > In contrast, for the CSS coding scheme, each layer’s neurons undergo a silent period of $L$ time steps (we set $L = 1$) before sending spikes to the next layer. As a result, the network latency for CSS is calculated as $T+N_{\mathit{layers}}$.

---

### Official Review · Reviewer_4dbK · 2024-11-01

**Soundness:** 3
**Presentation:** 1
**Contribution:** 2
**Rating:** 6
**Confidence:** 3

**Summary:**

The paper proposes a novel spiking input encoding called Canonic Signed Spike (CSS), which incorporates non-linearity into the encoding process by weighting spikes at each step of neural computation. To mitigate the temporal coupling phenomenon in the SNN using CSS encoding, the paper proposes a Ternary Self-Amplifying (TSA) neuron model and a one-step silent period, which reduces the encoding latency.

**Strengths:**

1. The idea that decouples the input and output when inference SNN is novel.
2. The experiments are detailed and convincing.

**Weaknesses:**

1. The method section is hard to read and some notations are not explained. For example, what are the detailed steps that generate Eq(5), and what is the meaning of τ (\tau) in Eq(5)?
2. The compared SOTA work in rate code is not superior. Fast-SNN [1] and Offset[2] works can achieve lossless conversion accuracy with less latency (time-step).
3. The energy consumption of the SNN encoded by CSS is missing.
I will raise my rating if the authors can solve the questions.

[1] Hu Y, Zheng Q, Jiang X, et al. Fast-SNN: fast spiking neural network by converting quantized ANN[J]. IEEE Transactions on Pattern Analysis and Machine Intelligence, 2023.
[2] Hao Z, Ding J, Bu T, et al. Bridging the gap between anns and snns by calibrating offset spikes[J]. arXiv preprint arXiv:2302.10685, 2023.

**Questions:**

1. Can the author provide a comparison between this work and other SOTA SNN conversion works such as (Fast-SNN[1], Offset[2])?
2. Can the author provide the number of synaptic operations of the SNN using CSS? The calculation code can be found in https://github.com/zhouchenlin2096/Spikingformer/tree/master/energy_consumption_calculation.
3. Can the authors provide a more simple explanation of the equivalence of ANNs and SNNs?


[1] Hu Y, Zheng Q, Jiang X, et al. Fast-SNN: fast spiking neural network by converting quantized ANN[J]. IEEE Transactions on Pattern Analysis and Machine Intelligence, 2023.
[2] Hao Z, Ding J, Bu T, et al. Bridging the gap between anns and snns by calibrating offset spikes[J]. arXiv prepri

---

> ### Author Response · Authors · 2024-11-25
>
> Thank you for your constructive feedback. Based on your suggestions, we have added two experiments, which are detailed in our public comment. Furthermore, we have enhanced the paper's structure by adding a preliminary section that outlines the fundamental principles of ANN-to-SNN conversion, emphasizing the critical roles of encoding methods and neuron models. Additionally, we have provided detailed proofs for the theorems and revised the summation indices in the equations to ensure clarity and avoid potential confusion. Below, we address your concerns in detail.
>
> * Can the author provide a comparison between this work and other SOTA SNN conversion works such as (Fast-SNN[1], Offset[2])?
>
> Yes, we have included comparisons with these two works. The two compared works are based on rate coding and rely on low-precision ANNs (for Offset, it specifically depends on the QCFS activation function, where output quantization is a key feature of this activation function). All three methods were tested by converting the same low-precision ANN to better evaluate the differences introduced by their respective encoding schemes.
>
> From the experimental results, our method achieves optimal performance with fewer time steps compared to Fast-SNN, which highlights the inherent advantage of spike weighting. Offset performs relatively well at 1 or 2 time steps but at the cost of significantly increased latency. Additionally, the calibrating effect of Offset appears to be limited, as its accuracy is surpassed by ours when the time steps increase to around 3. For more detailed information, please refer to our public comment.
>
> * Can the author provide the number of synaptic operations of the SNN using CSS? The calculation code can be found [here]
>
> Yes, we conducted experiments using the code you provided, and the results have been included in our public comment. Although our encoding scheme is not inherently sparse, it benefits from a significantly compressed number of time steps, which contributes to its energy efficiency.
>
> We first compared our approach with the pre-conversion ANN, where we observed a reduction in energy consumption by approximately fivefold. Unfortunately, we could not find the code for the TTFS-based approach for a direct comparison. As a result, we conservatively estimated the energy consumption of TTFS encoding based on relevant works, as detailed in our official comment. The result shows that by minimizing the number of time steps, CSS can achieve even lower energy consumption than TTFS (a 10% reduction). Finally, we selected Fast-SNN as a strong baseline for rate coding. Our results demonstrated that our encoding scheme reduces energy consumption by over 30%, while maintaining competitive accuracy.
>
> * Can the authors provide a more simple explanation of the equivalence of ANNs and SNNs?
>
> We've now incorporated the basic concept of ANN-SNN conversion in the Preliminary section. Below, we briefly explain the principle. The connection between ANN and SNN lies in the mapping of activation values to spike sequences. Through the encoding method, we can represent a spike sequence as a value equivalent to the activation value in the ANN.
>
> To be more specific, please refer to Eq. (5) in the newly uploaded PDF. By interpreting the spike sequence through rate coding and defining the "spike frequency" as $r_i^l[T]\coloneqq\frac{\sum_{t=1}^T S_i^i[t]}{T}$, Eq. (5) can be rewritten as:
> $$r_i^l[T]=\sum_j w_{ij}^{l}r_j^{l-1}[T]+\sum_{t=1}^T\frac{b_i^l}{T}-\frac{u_i^l[T]}{T}$$
> When $T$ is sufficiently large, we can ignore the last term on the right-hand side. Given that $r_i^l[T] \geq 0$, this simplifies to:
> $$r_i^l[T]=\max\left(\sum_{j}w_{ij}^{l}r_j^{l-1}[T]+\sum_{t=1}^{T}\frac{b_{i}^{l}}{T},0\right)$$
> Comparing this to the forward propagation process in an ANN:
> $$a_i^l=\max\left(\sum_{j}\hat w_{ij}^{l}a_j^{l-1}+\hat b_{i}^{l},0\right)$$
> we observe that equivalence between the two types of networks can be achieved by directly using the ANN weights and scaling the bias linearly! The importance of the encoding method and the neuron model stems from the fact that: 1) we need to interpret the spike sequence using some encoding scheme (here, rate coding), and 2) Eq. (5) is derived using some neuron model (here, IF model). In our work, it is precisely through 1) designing an appropriate encoding method and 2) developing a suitable neuron model for neural computation that we address the issues present in existing encoding schemes.

---

> > ### Comment · Reviewer_4dbK · 2024-11-26
> >
> > Thanks for the authors' reply. My questions are solved, I raise my score from 5 to 6.

---

### Official Review · Reviewer_ADde · 2024-11-01

**Soundness:** 2
**Presentation:** 2
**Contribution:** 2
**Rating:** 6
**Confidence:** 4

**Summary:**

This paper introduces a novel Canonical Signed Spike coding for Spiking Neural Networks to enhance performance and reduce latency in ANN-SNN conversions. The CSS incorporates non-linearity by weighting spikes and negative spikes, allowing more information per time step. The authors also propose a Ternary Self-Amplifying neuron model to further reduce latency. Experimental results on CIFAR-10 and ImageNet show that CSS improves inference latency and accuracy with minimal conversion loss, advancing SNN performance on large-scale datasets.

**Strengths:**

The paper's approach of using negative spikes for information encoding is innovative. This technique may compensate for the over firing issue and reduce conversion error.

The one-step silent period seems to be an effective approach which significantly enhances the performance of SNN models.

**Weaknesses:**

The ablation study provided in this paper is insufficient. As it stands, the empirical results primarily support the effectiveness of the negative spike and silent period techniques. A more detailed evaluation of the amplification factor's effect is necessary. Moreover, a comprehensive comparison between the TSA neuron and the standard IF neuron would better highlight the advantages of the proposed neuron model.

The paper is hard to follow and sometimes confusing. For example, the final neuron function for TSA neurons is not provided in the manuscript, which makes it difficult to fully understand the general behavior of the TSA neuron. Clear and detailed descriptions of such core elements are essential for readability and comprehension.

**Questions:**

Equation 3 makes no sense since the dirac function is a generalized function and is only well-defined within an integral over a specified range. The authors should consider using heaviside step function to describe the neuron firing.

Equation 3 appears to represent the discrete form of a LIF neuron function. However, based on the experiments, the scaling factor $\beta$ is consistently set to values greater than one, which is atypical for a traditional LIF neuron model.

To improve clarity, it would be beneficial if the authors provide the explicit mathematical function for the TSA neuron model.

It confuses me whether CSS coding serves as an input encoding method for the SNN or if it refers to the encoding capabilities inherent in the TSA neuron itself.

In lines 305-306, it is mentioned that a new term is added to the right side of Equation 8 after a silent period. Could the authors clarify the purpose of doing so?

It would be better if the authors can provide the pseudo code for the conversion and the inference process.

---

> ### Author Response · Authors · 2024-11-25
>
> Thank you for your valuable suggestions. We are delighted that you recognized the innovation in our approach. Indeed, the initial manuscript lacked smooth logic and clear descriptions, which posed challenges to truly understanding our method. Based on your advice, we have reviewed and revised the mathematical expressions and added the pseudocode for the TSA forward propagation process in the main text (while keeping the mathematical formulation in the appendix). We have uploaded a fully revised PDF for your reference. Additionally, we conducted experiments to study the impact of $\beta$, and the results have been included in the public comment. These results validate our analysis, and you can find more detailed information there. Below, we will address your concerns to the best of our ability.
>
> * Equation 3 makes no sense since the dirac function is a generalized function and is only well-defined within an integral over a specified range. The authors should consider using heaviside step function to describe the neuron firing.
>
> Indeed, we have replaced all parentheses with square brackets to emphasize the discrete nature of the domain. Specifically, $\delta[\cdot]$ denotes a unit impulse (we've now provided the definition of $\delta[\cdot]$ directly in a footnote in the main text).
>
> * Equation 3 appears to represent the discrete form of a LIF neuron function. However, based on the experiments, the scaling factor is consistently set to values greater than one, which is atypical for a traditional LIF neuron model. To improve clarity, it would be beneficial if the authors provide the explicit mathematical function for the TSA neuron model.
>
> Thank you for your suggestion. We have currently included the pseudocode for TSA forward propagation in the main text, as we believe it provides a more intuitive explanation, while the mathematical description remains in the appendix.
>
> We have now explicitly clarified that $\beta>1$ when introducing $\beta$. While settings like $\beta=p$ (where $p$ is a positive value) and $\beta = \frac{1}{p}$ can both theoretically ensure a weight difference of $p$ times between adjacent spikes, only $\beta>1$ is in practice feasible. To illustrate this, let $T$ represent the number of time steps and $\theta^l$ denote the spike amplitude. After weighting, the maximum value that the sequence can encode is $\sum_{t=0}^{T-1}\beta^t\theta^l$. Therefore, for encoding the same range, a smaller $\theta^l$ is achievable when $\beta>1$. Since $\theta^l$ determines the quantization error during encoding (as it represents the minimum value that a single spike can express), minimizing $\theta^l$ is essential. As a result, the proposed weighted scheme cannot be implemented using LIF neurons.
>
> * It confuses me whether CSS coding serves as an input encoding method for the SNN or if it refers to the encoding capabilities inherent in the TSA neuron itself.
>
> It refers to the encoding capabilities inherent in the TSA neuron itself. Input encoding is performed by the TSA neuron, as defined and proven in Theorem 2.
>
> * In lines 305-306, it is mentioned that a new term is added to the right side of Equation 8 after a silent period. Could the authors clarify the purpose of doing so?
>
> The reason we made this adjustment is to make it easier to satisfy Equation (8) (now Equation (10) in the revised version). In Corollary 1., we proved that meeting this condition ensures the accuracy of the converted SNN. In the revised version, we have added textual explanations following the theorems, clarifying how they guide the design of our method. We believe the updated version facilitates a smoother and more intuitive understanding.
>
> * It would be better if the authors can provide the pseudo code for the conversion and the inference process.
>
> That’s correct. We have now included the pseudocode for the conversion process in the appendix. The inference process essentially corresponds to the layer-by-layer propagation of spikes through TSA after conversion, as outlined in the TSA pseudocode (or alternatively, you may refer to the mathematical description of TSA).

---

### Official Review · Reviewer_qUNB · 2024-11-02

**Soundness:** 3
**Presentation:** 3
**Contribution:** 3
**Rating:** 6
**Confidence:** 5

**Summary:**

This paper introduces a novel approach that reduces coding time steps by weighting individual spikes, allowing each spike to transmit increased information density. The introduction of negative spikes successfully breaks temporal dependencies, thereby reducing inference delay. The proposed Compressed Spike Sequence (CSS) encoding method improves both throughput and processing speed in converted Spiking Neural Networks (SNNs), while maintaining minimal conversion accuracy loss, providing an effective solution for enhancing the practical efficiency of SNNs.

**Strengths:**

1.This paper proposes a novel non-linear coding approach named Compressed Spike Sequence (CSS) that breaks the linear relationship constraints inherent in traditional coding schemes.
2.The paper introduces negative spikes and a one-step silent period mechanism that balances coding efficiency and inference latency.
3.The method is comprehensively validated on standard datasets including CIFAR-10 and ImageNet, with extensive testing across network architectures of varying depths (such as VGG and ResNet), and detailed ablation studies analyzing the contribution of each component.
4.The proposed approach achieves nearly lossless ANN-to-SNN conversion with fewer time steps, and can be directly applied to full-precision ANNs without additional quantization training.

**Weaknesses:**

1.The proposed coding scheme lacks sparsity characteristics, which may lead to higher energy consumption.
2.The selection of parameters (such as \beta and \theta) lacks theoretical guidance.
3.Hardware implementation requires processing exponential operations of \beta, handling both positive and negative spikes, and controlling the silent period.

**Questions:**

1.What is the energy consumption of this SNN compared to ANN? It is suggested to add comparative experiments on energy consumption.
2.What is the level of support for this type of neuron in existing neuromorphic hardware? Is deployment and implementation feasible in the future?

---

> ### Author Response · Authors · 2024-11-25
>
> Thank you for your precise and valuable feedback. We are glad that you recognize the significance of our work, and we will address your questions below.
>
> * What is the energy consumption of this SNN compared to ANN? It is suggested to add comparative experiments on energy consumption.
>
> Thank you for your suggestion. We have included this section of experiments in the public comment. The results show that, although our encoding scheme lacks sparsity, the significant reduction in time steps provides a clear energy advantage. For example, the energy consumption of the converted full-precision VGG-structured SNNs are more than five times lower than that of the corresponding ANNs.
>
> Additionally, we compared our method with other neural coding approaches. Compared to the state-of-the-art rate coding method (Fast-SNN), our approach achieves over a 30% reduction in energy consumption. We also evaluated our approach against TTFS coding. However, by further reducing the time steps in CSS (e.g., by converting a quantized ANN), we were able to achieve approximately 10% lower energy consumption than TTFS.
>
> Overall, our coding scheme provides substantial energy savings while avoiding the large output latency typically associated with TTFS-based approaches.
>
> * What is the level of support for this type of neuron in existing neuromorphic hardware? Is deployment and implementation feasible in the future?
>
> Thank you for your questions. We are currently focused on the design and implementation of dedicated hardware. We will release the hardware design once the validation phase is successful. Our design builds upon existing neuromorphic computing platforms, ensuring high generality and flexibility.
>
> Specifically, the amplification coefficient $\beta$ can be efficiently implemented using shift operations. Given that the silent period for each layer is predetermined, a simple control signal is used to determine when to fire a spike. Additionally, we have developed specialized encoding methods to differentiate between positive and negative spikes.

---

### Author Response · Authors · 2024-11-28

## Further Revisions to Manuscript

__Dear Reviewers__,

Thank you for your timely follow-up and thoughtful feedback over the past few days. We sincerely appreciate the time and effort you have invested in reviewing our manuscript.

In response to your valuable suggestions, we have further made several revisions to the paper, summarized as follows:

### **Enhancements for Readability**
1. **Additional Figures**: We have added two figures to the main text. One illustrates the information content of spikes at different time steps after weighting, and the other visually demonstrates the phenomenon of temporal coupling of weighted spikes.
2. **Optimized Method Presentation**: Building on the previous version, we have further refined the presentation of the theorems for clarity. Specifically, we removed _Corollary 1_ and upgraded the original _Lemma 1_ to _Theorem 1_, making the approach in the _Methods_ section more direct. Below is a brief summary for the presentation of the _Methods_ section:
   - **Observation 1** focuses the discussion on controlling the residual membrane potential.
   - **Theorem 1** outlines all the factors affecting the residual membrane potential, guiding the introduction of negative spikes and TSA neurons.
   - With the encoding scheme and neurons clearly defined, **Theorems 2 and 3** establish the connection between CSS-SNN and ANN.

### **Additional Experimental Content**
In response to the feedback during the discussion, we have added the following experiments to the manuscript:
1. The impact of $\beta$ on network performance
2. A comparison with SOTA methods (Fast-SNN, Offset)
3. Energy analysis

Due to space limitations, we _have moved the ablation study on the silent period to the appendix_. However, given the significant role of negative spikes in CSS coding, the ablation study on negative spikes remains in the main text.

---

Overall, these updates integrate the insights from our discussions and have, we believe, significantly improve the quality of the manuscript. We kindly invite you to focus on the sections that interest you most.

Finally, should there be any remaining concerns or questions, please don’t hesitate to reach out. We are happy to address any further issues or provide clarification as needed.

---

### Meta-Review · Area_Chair_ekMr · 2024-12-22

**Metareview:**

This paper combines Canonic Signed Spike (CSS) coding scheme with a Ternary Self-Amplifying (TSA) neuron model to mitigate the relatively high time latency problem of ANN-SNN Conversion. Reviewers acknowledge the authors' detailed analysis about CSS coding, but express concerns about the hardware implementation and energy consumption of the converted SNNs in the inference stage. After the discussion period, three reviewers rate 6 and one reviewer rates 3. None of the reviewers strongly argue for acceptance. Therefore, the final decision is not to accept the paper.

**Additional Comments On Reviewer Discussion:**

Reviewer qUNB and DPaT express concern about whether the negative spike and silent period can be efficiently handled on neuromorphic hardware, while Reviewer ADdE and 4dbK point out that this work needs more detailed notation explanation and comparative experiments. The authors further provide supplementary discussion and experiments during the response phase. Overall, this paper would benefit from further clarification and analysis about its computational overhead and feasibility in hardware deployment.

---

### Decision · Program_Chairs · 2025-01-22

Reject